# Chromatin state origins of uterine leiomyoma

Maritta Räisänen [1,2], Eevi Kaasinen [1,2], Maija Jäntti[1,2], Aurora Taira [1,2], Emma Siili[3], Ralf Bützow[2,3], Oskari Heikinheimo [4], Annukka Pasanen[3], Auli Karhu [1,2], FinnGen*, Davide G. Berta[1,2], Niko Välimäki [1,2] & Lauri A. Aaltonen [1,2,5] ✉

Aberrations in the regulatory genome play a pivotal role in population-level disease predisposition. Annotation of the regulatory regions using appropriate primary tissues - instead of cell lines affected by selection and other confounding factors - could shed new light into mechanisms underlying common conditions. We test this approach in uterine leiomyomas, highly prevalent benign neoplasms of the myometrium, by creating 15-state chromatin annotations for myometrium and uterine leiomyomas. Integration with RNA-seq, ATAC-seq, HiChIP and methylation data enables us to compare the epigenomes of myometrium and ULs with distinct driver mutations, highlighting the role of bivalent regions in the neoplastic process. Subsequently, a genome wide association study meta-analysis is performed, using three different cohorts. Disease association loci are enriched at active chromatin, especially at enhancers, and harbor tumor- and driver mutation-specific chromatin states. At *SATB2* locus we show the effect of the risk genotype already in the normal tissue. Integration of genome-wide association studies and deep regulatory genomics data from the correct tissue type represents a powerful approach in understanding population-level disease predisposition.

Epigenetic modifications and their contribution to tumorigenesis has been a strongly emerging theme in cancer research. Recently, non-mutational epigenetic reprogramming was added as an enabling characteristic in the classical hallmarks of cancer[1]. A recent study even proposed that transient inactivation of Polycomb group components in *Drosophila* leads to epigenomic changes and tumorigenesis without contribution of somatic mutations[2], highlighting the importance of epigenetic mechanisms in tumorigenesis.

Chromatin structure is a key player in the functions of genomic loci. Different combinations of histone post-translational modifications (PTMs) can be used to determine chromatin states. For example, 15 chromatin states can be annotated based on the presence or absence of five histone PTMs. These different states describe different activities of the chromatin segment in question, marking for example, active transcription start sites (TSSs) and heterochromatin. The 15 states have been determined for many cell lines and a few tissue types[3], but not for myometrium. As cell lines are grown in an artificial environment and are affected by selection and other confounding factors, these are not the best models for regulatory genomics studies in the primary tissues. Because the impact of the regulatory genome in myometrial functions and tumorigenesis has become an important focus of study, there is a burning need for data on chromatin states in this tissue type.

[1]Department of Medical and Clinical Genetics, University of Helsinki, Helsinki, Finland. [2]Applied Tumor Genomics Research Program, Research Programs Unit, University of Helsinki, Helsinki, Finland. [3]Department of Pathology, University of Helsinki and Helsinki University Hospital, Helsinki, Finland. [4]Department of Obstetrics and Gynecology, University of Helsinki and Helsinki University Hospital, Helsinki, Finland. [5]iCAN Digital Precision Cancer Medicine Flagship, University of Helsinki, Helsinki, Finland. *A list of authors and their affiliations appears at the end of the paper. ✉e-mail: lauri.aaltonen@helsinki.fi

Myometrium refers to the smooth muscle wall of the uterus. It undergoes regular changes during the menstrual cycle and pregnancy, requiring dynamic control of gene expression. Bivalent promoters are marked by both active (H3K4me3) and repressive (H3K27me3) histone modifications, and removing the other creates a rapid way of switching on/off key developmental genes in a tissue-specific manner. Thus, bivalent promoters could play an important role in the regulation of the cyclic changes taking place in myometrium[4]. Uterine leiomyoma (UL) is the most common clinically significant neoplasia in women of reproductive age, affecting approximately 70% of women during their lifetime[5]. Approximately 25% of patients develop symptoms ranging from abnormal uterine bleeding to subfertility. Currently, the only curative treatment is surgical, and the disease causes a major burden to the affected individuals as well as society. ULs can be divided into mutually exclusive subclasses based on the genetic driver. Each driver presents a distinct model of UL genesis, and studying different subclasses provides valuable insight into the different mechanisms leading to UL genesis. The most common alteration is a mutation in the *MED12* gene comprising over 70% of ULs[6]. Other alterations include overexpression of *HMGA2*[7], *HMGA1,* and *PLAG1*, biallelic loss of *FH*[8], deficient H2A.Z loading by mutations in SRCAP complex genes[4], defects in Cullin 3-RING E3 ligase neddylation[9] and deletions at the *COL4A5/6* -locus[7].

Changes in specific histone PTMs and genomic regions between myometrium and UL have been reported[10,11]. In the study by Leistico et al. [10] three active histone modifications (H3K4me1, H3K4me3, and H3K27ac) were analyzed, and the signal from each of these separately and combined was able to separate myometrium from UL tissue. Changes in the active enhancer regions have been established previously[10,11], but the complete regulatory genome annotation of myometrium and ULs is still lacking. ULs in previous studies mostly comprise *MED12* mutant tumors, thus the differences in histone PTMs between UL subclasses arising from distinct genetic drivers are also not known.

In this study, we create 15-state chromatin annotations for myometrium and three UL subclasses, utilizing fresh frozen tissue samples from multiple individuals. The annotations are integrated with previously created gene expression, chromatin accessibility, 3D interactions, and methylation data from myometrium and UL subclasses. These data create a much improved understanding of the regulatory genome in myometrium, and the structure of the hereditary predisposition loci identified in genome-wide association studies (GWAS). These data also demonstrate epigenetic changes occurring during genesis of UL in a genetic subclass-specific manner.

## Results

### Chromatin state annotation of myometrium and UL subclasses

With chromHMM[12] we created 15-state chromatin annotations (Supplementary Fig. 1a) using five histone PTMs (H3K4me1, H3K4me3, H3K9me3, H3K27me3 and H3K36me3) for myometrium and three UL subclasses, MED12, HMGA2, and FH. We used three ULs from each subclass and myometrium samples from three UL patients. Myometrium tissues were collected from UL patients after hysterectomy, from sites not adjacent to tumors. We have previously created a 5-state model for myometrium[4] and with the addition of three markers (H3K4me1, H3K9me3, and H3K36me3) we were able to expand the model to 15 chromatin states. The chromatin states have not previously been assigned for ULs, and to study the distinct genetic drivers, we created the annotations for the most common subclasses MED12 and HMGA2, and for FH, which causes metabolic stress leading to UL genesis[13]. In addition to the subclass annotations, we created annotations for individual samples (3 myometria, 3 FH ULs, 2 HMGA2 ULs and 2 MED12 ULs, see Supplementary Fig. 1c). Together with previously created data layers (Fig. 1a) from the same molecular subclasses, we were able to thoroughly map the regulatory genome of myometrium and ULs with distinct genetic drivers.

In our previous work, we created a 5-state chromatin annotation for myometrium tissue[4]. Comparing that model and the current 15-state model, the expected concordance between corresponding states can be seen (Fig. 1b). All 15-state bivalent annotations (TssBiv, BivFlnk, EnhBiv) were most similar to the 5-state TssBiv-annotation, whereas the 5-state model annotation Other Active could be more precisely annotated as, for example, enhancers (Enh), genic enhancers (EnhG), transcribed regions (TxWk) and flanking active transcription start sites (TssAFlnk). With the addition of three histone markers in the 15-state model, we were able to annotate many previously quiescent chromatin regions as heterochromatin, ZNF/Repeats, enhancers and other active annotations in the 15-state model, compared to the previous 5-state model. The additional histone markers give us a more precise picture of the chromatin states in myometrium compared to the previous 5-state model, and extending the annotations into UL subclasses provides us with insights into the commonalities and differences occurring during UL genesis with different genetic drivers.

We performed hierarchical clustering for the individual samples to see if myometrium and the different UL subclasses differ based on chromatin segmentation data. With unsupervised hierarchical clustering, all individual samples and subclass segmentations are clustered based on their identity (Fig. 1c), confirming that chromatin segmentations differ between myometrium and UL subclasses genome-wide. Relative contribution of each annotation was shown to be quite similar between all UL subclasses and myometrium (Fig. 1d), and this was seen also in segmentations created for each individual myometrium and UL sample (Supplementary Fig. 1c). We compared the contribution of each of the 15 annotations between myometrium and ULs in our sample set. Interestingly, the proportion of heterochromatin tended to be lower in the three myometrium samples compared to ULs (Welch Two Sample t-test, adj. P = 0.099). As expected, the most common annotation in all groups was quiescent chromatin, comprising 53% of the whole genome in myometrium (Fig. 1d).

Heterochromatin is marked only by H3K9me3, and this state alone separated myometrium and all three UL subclasses (Supplementary Fig. 1d). To confirm that the difference seen between the groups was not just technical variation in the H3K9me3 ChIP-seq data intensities, we compared fragment counts at called peaks between paired myometrium and UL samples, and confirmed that most of the H3K9me3 peaks contained similar amounts of fragments in ULs and myometrium (Supplementary Fig. 1e). This notion held for all three myometrium-UL pairs analyzed. Interestingly, there were some H3K9me3 peaks containing more fragments in myometrium than in UL, in contrast to the genome-wide phenomenon where we saw more heterochromatin in all three UL subclasses. These peaks were located on 3'UTR regions of ZNF-genes.

In our previous work[4], we showed that differentially more accessible regions (DARs) in UL subclasses compared to myometrium were enriched at enhancer regions. Annotations used previously included tissue types from Roadmap Epigenomics, but not uterine tissues. With the created 15-state chromatin annotations from myometrium and UL subclasses, we now confirmed that DARs were enriched to enhancers also using annotations derived from the correct tissue type, validating our previous conclusions (Fig. 2a).

Similarly, we showed that hypermethylated regions were enriched at bivalent regions in MED12 and HMGA2 ULs, and other active chromatin in FH ULs in myometrium 5-state model for chromatin annotation[4]. With the 15-state annotation of myometrium and ULs, we confirmed the finding for MED12 and HMGA2 groups, and could pinpoint in FH tumors that the hypermethylated regions annotated as other active chromatin in the 5-state model were in fact enhancers in the 15-state model (Fig. 2b).

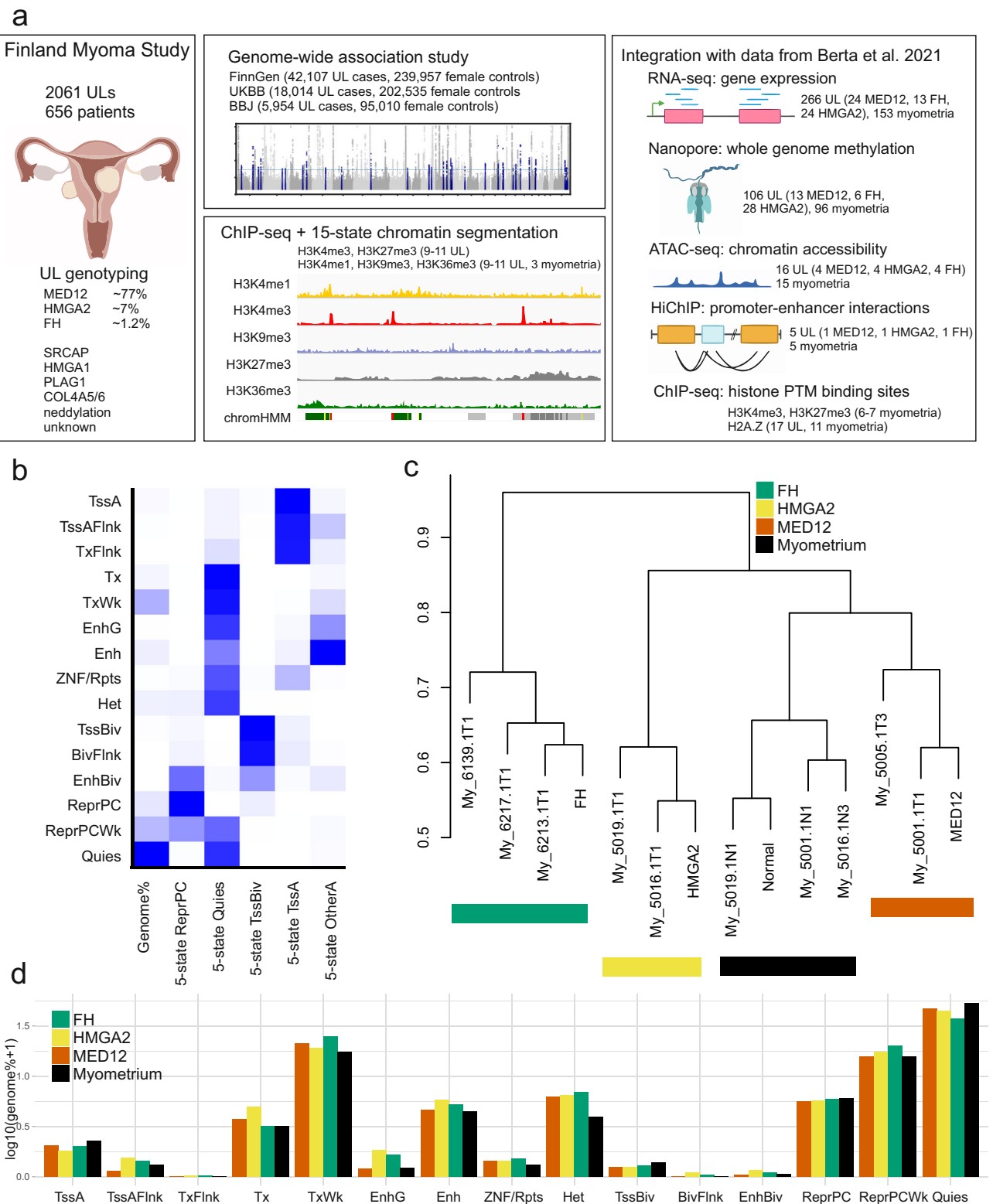

While DARs were also enriched at enhancers in all UL subclasses (Fig. 2a), enhancers with hypermethylated CpGs in FH ULs are distinct from those with more accessible chromatin (Supplementary Fig. 1b). To further examine the enhancers hypermethylated in FH ULs, we studied the differences in transcription factor motif accessibility at these regions between FH ULs and myometrium using chromVAR[14]. We found 127 motifs to be

differentially accessible between FH ULs and myometrium, including estrogen receptors 1 and 2 (Supplementary Data 2).

We have also shown previously[4] that H2A.Z binding sites in myometrium were strongly enriched to active transcription start sites and bivalent regions. With the 15-state model, enrichment to bivalent regions was even stronger than with the previously used 5-state model (Berta et al.[4] Supplementary Fig.).

**Fig. 1 | Chromatin landscape of myometrium and UL genetic subclasses.**
**a** Finland myoma study consists of 2061 ULs collected from 656 patients. The middle panel shows data created and analyzed in this study, and the right-hand panel depicts previously created data integrated to new data layers in this study. The schematic shows all data types utilized in this study, and shows how chromatin segmentations are created from ChIP-seq data with an example myometrium sample. Created in BioRender. Räisänen, M. (2025) https://BioRender.com/99cfkej. **b** 15-state chromatin annotation of myometrium correlates with the previous 5-state model and provides more specific annotations for the regulatory genome. A darker blue color corresponds to a greater fold enrichment for a column-specific coloring scale. ReprPC: Repressed by Polycomb; Quies: quiescent chromatin; TssBiv: Bivalent/poised TSS; TssA: Active TSS; OtherA: other active chromatin. **c** UL subclasses cluster separately based on chromatin annotations. **d** Genome-wide contribution of each of the 15-state annotations in myometrium and three UL subclasses. TssA Active TSS, TssAFlnk Flanking active TSS, TxFlnk Transcr. at gene 5′ and 3′, Tx Strong transcription, TxWk Weak transcription, EnhG Genic enhancers, Enh Enhancers, ZNF/Rpts ZNF genes + repeats, Het Heterochromatin, TssBiv Bivalent/poised TSS, BivFlnk Flanking bivalent TSS/Enh, EnhBiv Bivalent enhancer, ReprPC Repressed Polycomb, ReprPCWk Weak repressed Polycomb, Quies Quiescent/low. Source data are provided as a Source Data file.

## Myometrium bivalent regions are deregulated in UL

To assess if the location of the annotations changes between myometrium and ULs, we calculated the proportion of each UL annotation at myometrium chromatin state loci, providing us with the degree of change in each chromatin state between myometrium and ULs. For this analysis, the 15 annotations were combined into seven to examine larger-scale changes in chromatin annotations between the groups. In each UL subclass, bivalent regions showed the most change from myometrium annotation, having only 39-60% similarity with the myometrium annotation (Fig. 3a, b).

Bivalent regions (TssBiv, BivFlnk, EnhBiv) comprised 0.46% of the whole genome in myometrium, and 0.31-0.51% in UL subclasses. We annotated 2197 autosomal genes as bivalent in myometrium (TssBiv annotation within 1000 bp from transcription start site, Supplementary Data 3). When we performed hierarchical clustering for the expression of these genes, myometria and UL subclasses clustered separately (Fig. 3c).

Next we searched for genes annotated as bivalent in myometrium, but as either activated or repressed in ULs. Genes with activating marks in MED12 and HMGA2 ULs showed higher expression values in RNA-seq data (UL vs. myometrium), whereas repressed genes had lower expression in all ULs compared to myometrium (Fig. 3d). In FH ULs the mean expression of genes with activating marks was higher than in myometrium (mean log2 fold change = 0.18), but the effect was not as striking as in MED12 (mean log2 fold change = 0.36) and HMGA2 (mean log2 fold change = 0.33) ULs. One hundred and forty-eight bivalent genes were annotated with active TSS (TssA or TssAFlnk) and 16 with repressed polycomb (ReprPC or ReprPCWk) in all UL subclasses. All genes activated or repressed in ULs are shown in Supplementary Data 3.

When we looked at how regions of each annotation in ULs were annotated in myometrium, we saw the largest change in heterochromatin that was annotated as quiescent in myometrium (Supplementary Fig. 2), which reflected the result seen in the total amount of this annotation in ULs compared to myometrium (Fig. 1d). Interestingly, approximately 30% of UL bivalent regions were annotated as repressed by polycomb in myometrium. These data suggest that in addition to bivalent regions changing to active or repressed in UL, repressed regions in myometrium gain activating histone modifications, and thus, display bivalency in UL.

## More accessible UL enhancers change expression of nearby genes

To study the changes in enhancer activity emerging in different UL subclasses, we determined more accessible UL enhancers for each UL subclass with ATAC-seq data. Enhancers were annotated as more accessible, if there were no overlapping enhancers in the myometrium annotation and the enhancer was more accessible in UL subclass compared to myometrium (adj. P < 0.05; log2FC > 0). We annotated 4023 such enhancers in MED12, 2948 in HMGA2, and 8379 in FH subclass. Most of the more accessible UL enhancers were intronic or intergenic, and 134 of them were shared by all UL subclasses. Over 60% of the more accessible UL enhancers were annotated as quiescent chromatin in myometrium.

Mapping of target genes for enhancer elements is not straightforward, as the target is not always the closest gene, and enhancers can affect multiple targets. We used three different approaches to map the possible target genes for more accessible UL enhancers (Fig. 4a). First, we analyzed expression of the closest protein-coding genes expressed in myometrium and/or ULs, to get an overview of the expression changes possibly caused by the emerging more accessible enhancer elements in UL subclasses. Altogether, 1925 annotated closest genes were shared by at least 2/3 UL subclasses.

For comparison, we used ENCODEs rE2G enhancer-gene link data[15] from 352 cell and tissue types to map target genes for more accessible UL enhancers. We annotated 3616 target genes in at least 2/3 UL subclasses, and 1160 out of 1925 previously annotated closest genes were found also in the ENCODE enhancer-gene link data, providing confidence in the regulatory link between the more accessible UL enhancers and annotated closest target genes.

We have previously created HiChIP data against H3K27ac to map 3D promoter-enhancer interactions for ULs and matched myometria[4]. We used these data to map differential interactions between pooled ULs and myometrium with 5 kb bin size (DiffLoops). 32-41% of the more accessible enhancers were overlapping UL-specific HiChIP-links in each UL subclass. Next we searched for possible target genes for these enhancers on the other end of the differential interaction. Genes with TSS in the target bin of HiChIP DiffLoops enhancer bin containing a more accessible enhancer were curated and their RNA-seq data for differential expression was examined. With these data we saw that when there was an increased interaction in UL, the gene expression (UL subclass vs. myometrium) was elevated compared to the genes with an increased interaction in myometrium, as one would not expect to see any regulatory effects of the latter enhancers in UL (Fig. 4b). 285 genes had a more accessible enhancer linked to them by a UL-specific loop in at least two of the three UL subclasses.

We identified 64 target genes for the more accessible UL enhancers in at least two UL subclasses with all three aforementioned methods to map target genes (Fig. 4c, Supplementary Data 4). Sixty of these genes were differentially expressed (FDR < 0.05) in at least one UL subclass compared to myometrium, and UL subclasses and myometria clustered separately based on their expression (Fig. 4d). One of these genes is *SATB2*, which was the most significantly overexpressed gene in ULs. With HiChIP DiffLoops, we mapped the enhancer regulating *SATB2* expression in ULs to downstream of the gene (Fig. 4e). These interactions were present only in ULs. This downstream region also contained many more accessible regions in ULs compared to myometrium, highlighting the importance of this regulatory region in ULs (Supplementary Fig. 3).

## GWAS signals emerge from active chromatin

Next we examined inherited UL risk loci by meta-analysis combining three GWASs: FINNGEN, UK Biobank (UKB), and Biobank Japan (BBJ), providing a total of 66,075 UL cases and 537,502 female controls. We identified 149 loci with a genome-wide significant association to ULs (Supplementary Data 5), of which 35 loci have not previously been implicated in UL predisposition (Table 1, Fig. 5a).

With chromatin state annotations from myometrium and UL subclasses, GWAS signals map to active chromatin regions more than expected by chance, in particular at genic enhancers (Fig. 5b). To further characterize the GWAS loci (most significant SNP +2000bp flanks), we labeled these regions with myometrium and UL chromatin annotations. With this approach we can divide GWAS regions into four

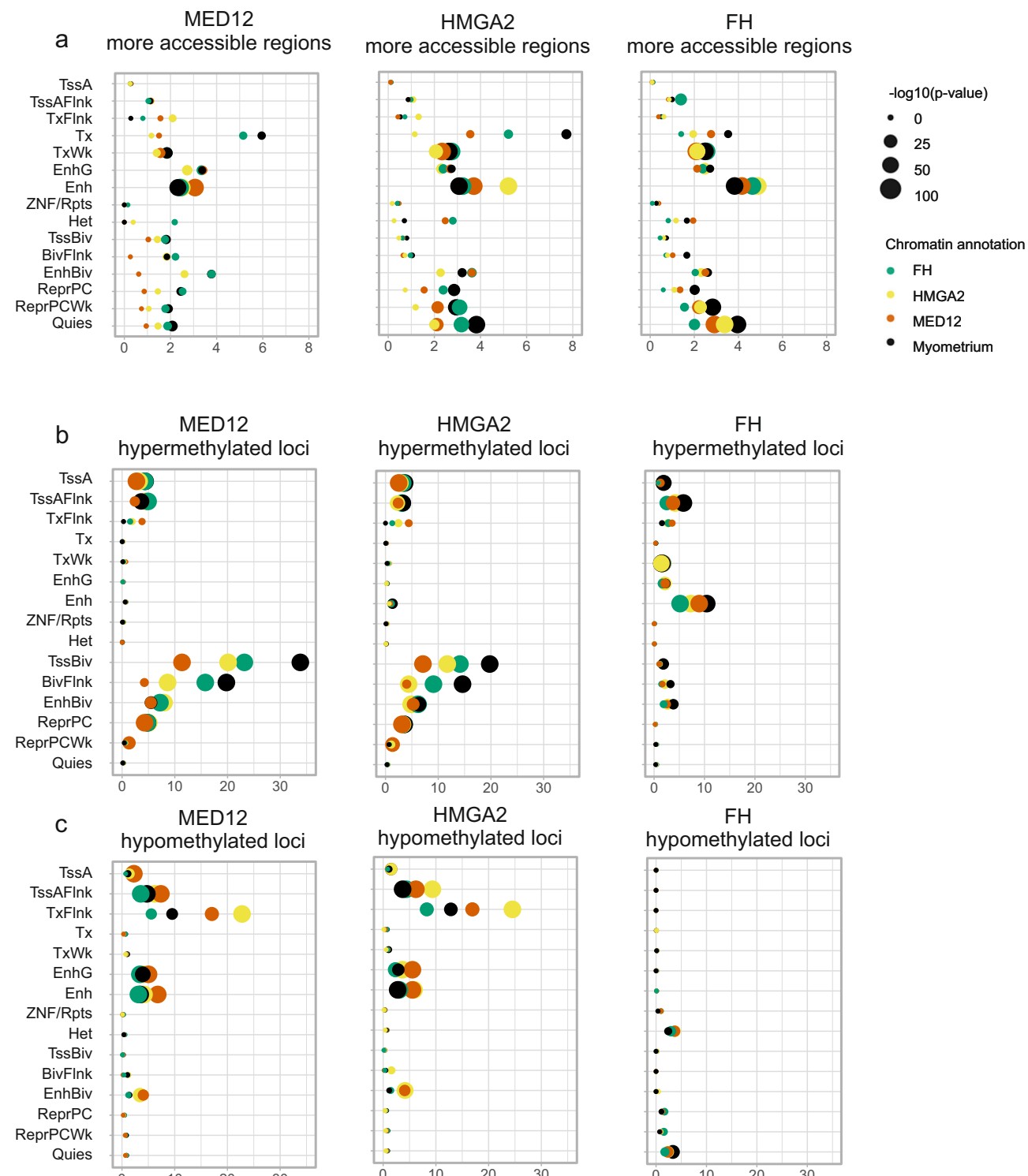

**Fig. 2 | 15-state chromatin annotation of myometrium and UL subclasses provides insight into biological changes seen in other data types. a** More accessible regions are enriched at enhancers (Enh) in all UL subclasses. All tested regions in differential analyses were used as background in enrichment analyses. **b** Hypermethylated loci in MED12 and HMGA2 are enriched at bivalent regions (TssBiv and BivFlnk), and at enhancers in FH ULs. **c** Hypomethylated regions in MED12 and HMGA2 ULs are enriched at active annotations (TssAFlnk, TxFlnk, Enh and EnhG). Statistics are calculated with Fisher's exact test implemented in the LOLA package. TssA Active TSS, TssAFlnk Flanking active TSS, TxFlnk Transcr. at gene 5' and 3', Tx Strong transcription, TxWk Weak transcription, EnhG Genic enhancers, Enh Enhancers, ZNF/Rpts ZNF genes + repeats, Het Heterochromatin, TssBiv Bivalent/poised TSS, BivFlnk Flanking bivalent TSS/Enh, EnhBiv Bivalent enhancer, ReprPC Repressed Polycomb, ReprPCWk Weak repressed Polycomb, Quies Quiescent/low. Source data are provided as a Source Data file.

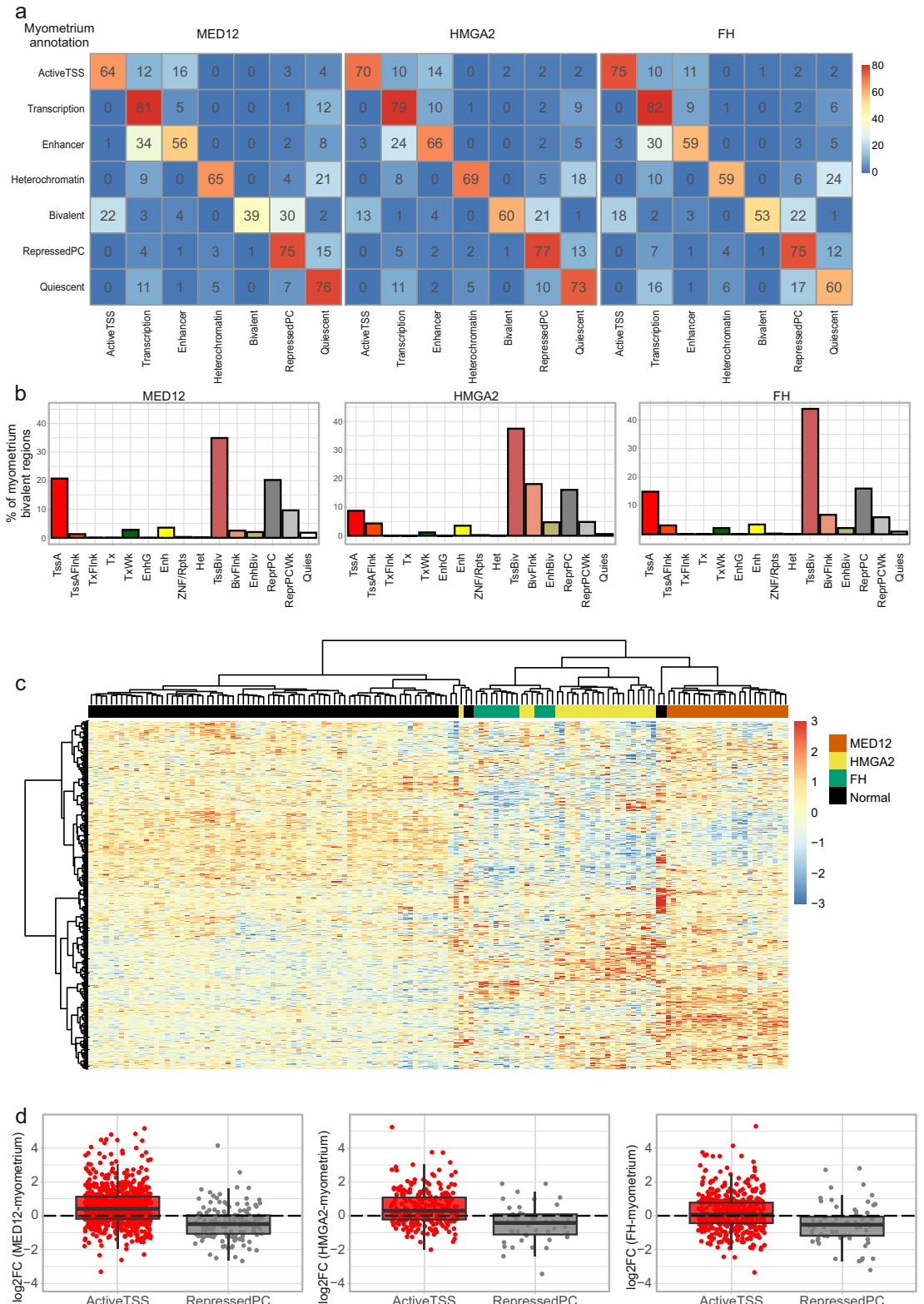

categories based on their chromatin annotations in myometrium: Transcription (53 regions), Enhancer and/or other active chromatin (37 regions), Repressed polycomb (20 regions) and Quiescent (29 regions) (Supplementary Fig. 4). In addition 10 regions mapping to chromosome X are mostly annotated with repressed polycomb annotations. We performed unsupervised hierarchical clustering based on chromatin annotations at these GWAS loci (most significant SNP +2000bp

flanks), separating myometrium and UL subclasses from each other (Fig. 5c).

With chromatin annotations for myometrium and UL subclasses MED12, HMGA2 and FH we could characterize regions that have different annotations in myometrium and ULs. One interesting example of a differentially annotated GWAS locus was an intronic region of *IGF2BP2*, which was annotated as active chromatin in HMGA2 and FH

**Fig. 3 | Bivalent regions are deregulated in ULs. a** Bivalent regions change the most from myometrium in UL annotation. Each square shows the percentage of the myometrium annotation with the respective UL annotation shown in x-axis. **b** 15-state annotations of myometrium bivalent chromatin (TssBiv, BivFlnk, EnhBiv) in UL subclasses. **c** Heatmap of the expression of myometrium bivalent genes (n = 2087) in myometrium and MED12, HMGA2, and FH ULs. **d** Myometrium bivalent genes with a chromatin state change to activated (TssA, TssAFlnk) or repressed (ReprPC, ReprPCWk) chromatin in ULs show differences also on the gene expression level. Centre line, median; box limits, 25% and 75% quartiles; whiskers, 1.5 × interquartile range (IQR) past the quartiles. MED12 ActiveTSS n = 781, ReprPC

n = 155 genes. HMGA2 ActiveTSS n = 267, RepressedPC n = 47 genes. FH Active TSS n = 463, ReprPC n = 69 genes. Gene expression log2 fold change calculated previously using 38 MED12, 44 HMGA2 and 15 FH ULs, and 162 myometrium biological replicates. TssA Active TSS, TssAFlnk Flanking active TSS, TxFlnk Transcr. at gene 5' and 3', Tx Strong transcription, TxWk Weak transcription, EnhG Genic enhancers, Enh Enhancers, ZNF/Rpts ZNF genes + repeats, Het Heterochromatin, TssBiv: Bivalent/poised TSS, BivFlnk Flanking bivalent TSS/Enh, EnhBiv Bivalent enhancer, ReprPC Repressed Polycomb, ReprPCWk Weak repressed Polycomb, Quies Quiescent/low. Source data are provided as a Source Data file.

ULs, but as repressed polycomb and quiescent chromatin in myometrium and MED12 ULs (Fig. 5d). *IGF2BP2* is overexpressed in HMGA2 and FH subclasses, and has previously been identified as a target gene of HMGA2[16]. We hypothesized that the *IGF2BP2* GWAS risk allele is more common among patients with HMGA2 ULs compared to other patients, but the difference was not significant (AF = 0.31 versus 0.28; OR = 1.15; P = 0.5).

One of the loci with genome-wide significant association with UL was identified downstream of the *SATB2* gene (Fig. 4e), which has previously been linked to somatic UL genesis[4,17]. This region was annotated as quiescent chromatin in myometrium but active elements including enhancers and transcribed chromatin emerged in all UL subclass annotations (Fig. 4e, Supplementary Fig. 3). The regulatory region contains DARs in all UL subclasses, and interestingly, we found that the chromatin accessibility of two of these was associated with the genotype of the *SATB2* risk allele rs7559104 already in myometrium tissue (Fig. 6). Homozygous risk-allele carriers had the most accessible chromatin at these loci, while the homozygous non-risk-allele carriers had less accessible chromatin (Fig. 6d, e). Both of these regions were connected to the *SATB2* gene promoter in HiChIP data from ULs (Fig. 4e), further supporting the functionality of these regions in UL genesis.

Functional mapping and annotation (FUMA)[18] of the GWAS SNPs identified 529 lead-SNPs, which the default FUMA analysis mapped to 849 different genes based on position (Supplementary Data 6). The resulting functional consequences of these SNPs were predominantly intronic (Supplementary Fig. 5). Gene expression analysis of these 849 genes across 54 tissue types (GTEx v8) displayed enrichment of up-regulated genes particularly in uterine tissue (Supplementary Fig. 6). An alternative tissue expression analysis (MAGMA)[19] of GWAS SNPs ranked uterine tissue as the seventh of the 53 tissue types analyzed (Supplementary Fig. 7). Further FUMA gene-set analysis indicated these 849 genes are enriched to biological processes involving DNA damage response, telomere organization and cell cycle (Supplementary Fig. 8). Other significantly (FDR adjusted P < 0.05) enriched pathways from the FUMA gene-set analysis are given in Supplementary Data 7. In RNA-seq data, hierarchical clustering of the 849 genes revealed distinct clusters for myometrium and each of the three molecular subclasses: MED12, FH, HMGA2 (Supplementary Fig. 9). The RNA-seq data revealed altogether 97 significant expression quantitative trait loci (eQTL) for myometrium and ULs (Supplementary Data 8). Taken together, these results give strong support for the tissue-specificity of the GWAS SNPs.

**Combined chromatin annotations and weaker GWAS signals identify loci with possible contributions to UL genesis**

We proceeded with the long tail of suggestive GWAS hits (P < 1e-5) to find candidate predisposition loci that also displays a change in chromatin status. We examined the GWAS regions as described above, but used P < 1e-5 as a significance threshold, and found altogether 458 regions (Supplementary Data 9), including the genome-wide significant loci described above. We mapped the more accessible UL enhancers at these regions, and found 60 regions where every UL subclass has a more accessible enhancer.

Next we annotated the most significant SNP+2000bp flanks of these regions to find differences between myometrium and ULs. We characterized regions with three annotations: Active (TssA, TssAFlnk, TxFlnk, Tx, TxWk, EnhG, Enh), Bivalent (TssBiv, BivFlnk, EnhBiv) and Repressed (ZNF/Rpts, Het, ReprPC, ReprPCWK, Quies). We searched for regions inactive in myometrium (100% of the bases with repressive annotations) and activated in ULs (>=20% of the bases with active annotation in all UL subclasses). We found 15 such regions, one mapping to the 3'-end of gene *BEND3* (one SNP, p = 8.3e-6) in which loss-of-function variants have previously been associated with UL predisposition[20].

## Discussion

Epigenetic changes often occur in and contribute to tumorigenesis, and the regulatory genome plays a pivotal role in population-level disease predisposition. In this study we created 15-state chromatin annotations for myometrium, and MED12, HMGA2, and FH uterine leiomyomas. Together with chromatin accessibility, gene expression, promoter-enhancer 3D interactions and methylation data from these same groups, we were able to show how the overall chromatin landscape changes in ULs compared to myometrium. Subsequently, we performed UL GWAS meta-analysis combining three cohorts, and were able to identify the regulatory features associated with predisposition loci, and show how the chromatin landscape at some of these loci changes during UL genesis. Taken together, these data create a thorough view of the regulatory genome landscape of myometrium and the three UL subclasses.

With the 15-state model for myometrium and ULs created in this work, we were able to validate previous findings and conclusions relying on annotations derived from other tissue types. Also, previous 5-state annotations from myometrium were upgraded; for example enhancers were identified within regions previously annotated as other active chromatin. The use of the correct tissue type for 15-state annotation resulted in much improved signals when integrated with ATAC-seq, H2A.Z ChIP-seq, and GWAS data, highlighting the value of the upgraded annotation.

Overall genome-wide contribution of each of the 15 annotations was similar between all analyzed groups, although most ULs were annotated with more heterochromatin than the myometrium samples. When examining the raw ChIP-seq data of H3K9me3, the histone PTM marking heterochromatin, we surprisingly noticed that some peaks have more H3K9me3 binding in myometrium than in ULs, especially at 3'UTRs of ZNF-genes. Interestingly, *ATRX*, a gene that is often mutated in uterine leiomyosarcoma[21], a malignant tumor of the myometrium, binds to the 3'exons of these genes to preserve H3K9me3 enrichment[22]. Most of the emerging heterochromatin regions in ULs are annotated as quiescent chromatin in myometrium, which could potentially mean that this difference is not linked to genomic landscape of this marker itself, but rather to the chromatin accessibility for the used antibody to this PTM; variation in the packaging of heterochromatin regions could possibly explain the seen difference.

Hypermethylated regions in FH ULs were enriched at enhancers (Fig. 2b). FH deficiency in these tumors leads to accumulation of fumarate, which has previously been shown to inhibit TET2 enzyme[23],

and cells with TET2 deficiency harbor hypermethylation at enhancer regions[24,25]. Interestingly, TET2 has been shown to co-bind with estrogen receptor (ER) at enhancers and is required for ER binding[26].

We showed significantly reduced accessibility at hypermethylated myometrial enhancers with ESR1 and ESR2 motifs in FH ULs compared to myometrium, unlike when the analysis was run with all myometrial

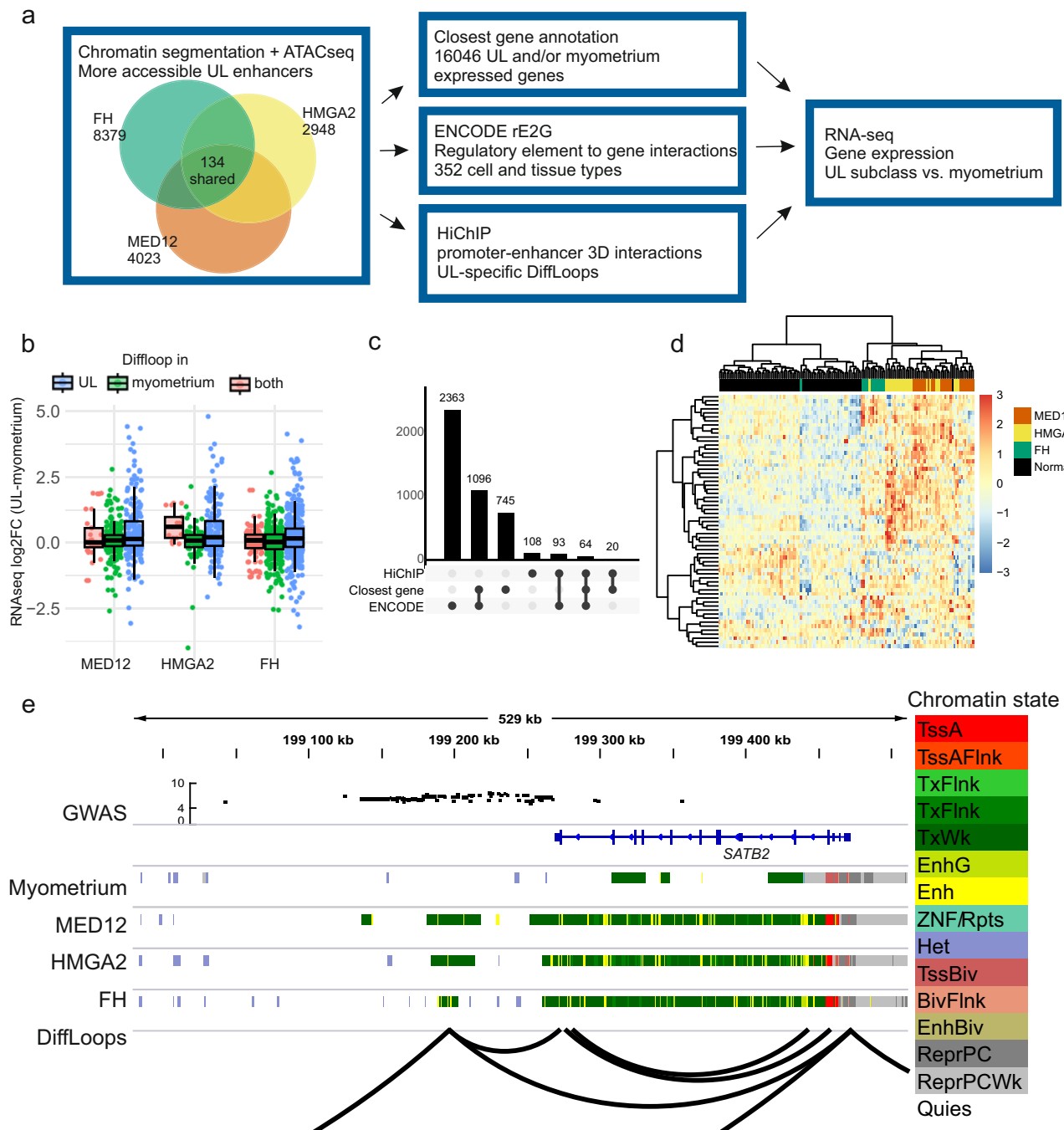

**Fig. 4 | Gene expression changes driven by more accessible UL enhancers.**
**a** Flow chart of target gene mapping of more accessible UL enhancers. We used three different methods to map target genes. Shared target genes are shared by all three UL subclasses. **b** Genes with UL-specific link to a more accessible enhancer tend to have higher gene expression than genes with a myometrium-specific link to a more accessible UL enhancer locus. Centre line, median; box limits, 25% and 75% quartiles; whiskers, 1.5 × interquartile range (IQR) past the quartiles. MED12 loop in UL n = 245, in myometrium n = 277, and in both n = 34 genes. HMGA2 loop in UL n = 163, in myometrium n = 77, and in both n = 17 genes. FH loop in UL n = 442, in myometrium n = 438 and in both n = 117 genes. Gene expression log2 fold change calculated previously using 38 MED12, 44 HMGA2, and 15 FH ULs, and 162 myometrium biological replicates. **c** Annotated target genes in at least 2 UL subclasses in each of the used methods. 64 genes were annotated as targets in at least 2/3 UL

subclasses with all three methods. **d** Gene expression of the 64 target genes in at least 2 UL subclasses as defined through all three methods divides myometrium and UL subclasses in hierarchical clustering. **e** *SATB2* regulatory region downstream of the gene links to TSS with HiChIP differential interactions in ULs. Figure shows GWAS SNPs associated with ULs (y-axis shows logarithmic p-value of the UL association of SNP), chromatin annotation for myometrium, MED12, HMGA2 and FH ULs and differential interactions between pooled ULs and myometrium. TssA Active TSS, TssAFlnk Flanking active TSS, TxFlnk Transcr. at gene 5' and 3', Tx Strong transcription, TxWk Weak transcription, EnhG Genic enhancers, Enh Enhancers, ZNF/Rpts ZNF genes + repeats, Het Heterochromatin, TssBiv Bivalent/poised TSS, BivFlnk: Flanking bivalent TSS/Enh, EnhBiv Bivalent enhancer, ReprPC Repressed Polycomb, ReprPCWk Weak repressed Polycomb, Quies Quiescent/low. Source data are provided as a Source Data file.

**Table 1 | Genome-wide significant GWAS loci associated with ULs**

| Locus | Plausible target | Chr. | Position | Ref. | Alt. | AF | OR | P | Annotation |
|---|---|---|---|---|---|---|---|---|---|
| rs28503425 | *TYMS* | 18 | 660647 | C | A | 0.16 | 0.93 | 3.2E-15 | RepressedPC |
| rs5742915 | *PML* † | 15 | 74044292 | T | C | 0.41 | 1.05 | 6.8E-15 | Transcription |
| rs1870940 | † | 1 | 155011887 | G | A | 0.24 | 1.07 | 1.1E-14 | Transcription |
| rs200315340 | *MYH11* † | 16 | 15737546 | C | T | 0.01 | 1.38 | 3.7E-14 | Transcription |
| rs28582771 | *MED12L* † | 3 | 150304137 | A | T | 0.40 | 0.95 | 5.1E-13 | Quiescent |
| rs79857163 | *MAP3K4* | 6 | 160986687 | C | T | 0.01 | 1.39 | 1.4E-12 | RepressedPC |
| rs56944467 | † | 19 | 814680 | G | T | 0.22 | 1.06 | 2.4E-11 | Transcription |
| rs141823469 | *WNK1,RAD52* † | 12 | 865142 | T | TC | 0.25 | 1.06 | 3.3E-11 | Transcription |
| rs752307 | *PARP1* | 1 | 226363828 | C | G | 0.76 | 1.05 | 4.5E-11 | Transcription |
| rs3794964 | *UBE2M* | 19 | 58559549 | T | C | 0.25 | 1.05 | 7.4E-11 | Enhancer |
| rs970394454 | *IGF2R* | 6 | 159827881 | A | G | 0.01 | 1.39 | 9.9E-11 | Enhancer |
| rs525704 | *MORF4L2* † | X | 103736086 | T | C | 0.54 | 1.05 | 1.4E-10 | RepressedPC |
| rs481548 | *PGR* | 11 | 101026780 | C | G | 0.77 | 0.94 | 2.3E-10 | Transcription |
| rs4641 | *LMNA* † | 1 | 156137743 | C | T | 0.18 | 1.05 | 2.8E-10 | Transcription |
| rs11603541 | *CCND1, FGF3/4/19* | 11 | 69657605 | C | G | 0.22 | 1.05 | 3.5E-10 | RepressedPC |
| rs10845387 | *ETV6* | 12 | 11604809 | G | A | 0.35 | 0.96 | 9.1E-10 | Transcription |
| rs1256059 | *ESR2* | 14 | 64243699 | A | G | 0.56 | 1.04 | 1.2E-09 | RepressedPC |
| rs35039038 | † | 15 | 75118713 | A | AC | 0.60 | 1.05 | 1.6E-09 | Transcription |
| rs11074422 | *GPRC5B* | 16 | 19988239 | A | T | 0.38 | 1.04 | 1.8E-09 | Quiescent |
| rs191988507 | *BECN1* | 17 | 42821278 | A | G | 0.03 | 1.15 | 2.1E-09 | Transcription |
| rs7903091 | *MICU1* † | 10 | 72381281 | T | C | 0.53 | 0.96 | 2.8E-09 | Transcription |
| rs76148823 | *CDKN2C* | 1 | 51189375 | A | C | 0.02 | 1.18 | 4.6E-09 | Quiescent |
| rs9826148 | *ZBTB20* † | 3 | 114746011 | C | T | 0.11 | 0.95 | 5.0E-09 | Transcription |
| rs11059824 | † | 12 | 128680506 | G | T | 0.41 | 1.04 | 1.1E-08 | Enhancer |
| rs6668749 | *MDM4,PIK3C2B* † | 1 | 204437293 | G | T | 0.23 | 1.04 | 1.4E-08 | Transcription |
| rs4244317 | *SORBS1* | 10 | 95540012 | T | A | 0.43 | 0.96 | 1.5E-08 | Enhancer |
| rs9686502 | *WDR41* † | 5 | 77327596 | T | G | 0.52 | 1.04 | 1.5E-08 | Enhancer |
| rs6608476 | † | X | 116050678 | A | G | 0.64 | 0.96 | 1.9E-08 | Heterochromatin |
| rs58649165 | *ANXA5* † | 4 | 121772597 | C | A | 0.40 | 1.04 | 2.1E-08 | RepressedPC |
| rs112833147 | † | X | 125949306 | C | CT | 0.57 | 1.05 | 2.6E-08 | RepressedPC |
| rs10962633 | *BNC2* | 9 | 16841974 | T | C | 0.25 | 1.05 | 2.7E-08 | Transcription |
| rs681998 | *SMRC8* | 17 | 18361135 | A | G | 0.40 | 1.03 | 2.7E-08 | Transcription |
| rs2384686 | *BRSK1* | 19 | 55326163 | G | A | 0.28 | 0.95 | 2.8E-08 | RepressedPC |
| rs7559104 | *SATB2* | 2 | 199224759 | G | T | 0.62 | 0.96 | 3.0E-08 | Quiescent |
| rs754058 | † | 2 | 171036607 | T | C | 0.68 | 1.03 | 4.0E-08 | Transcription |

† Multiple genes and/or no clear candidate target gene. Chr chromosome, Position genomic position in the hg38 reference, Ref reference allele, Alt alternative allele, AF alt. allele frequency in the FINNGEN cohort; OR odds-ratio for the alt. allele in the FINNGEN cohort; P: inverse-variance weighted fixed effects meta-analysis P-value for UL association (N = 66,075 UL cases and N = 537,502 female controls); Annotation: chromatin state at the risk locus in myometrium (Supplementary Fig. 4).

enhancers (Supplementary Data 2), showing that reduced accessibility of ESR1 and ESR2 motifs is specific for the hypermethylated enhancers. This finding is in line with the hypothesis that accumulating fumarate leads to TET2 inhibition and enhancer hypermethylation in FH deficient ULs, and suggests that FH mutant lesions may be less dependent on hormonal cues than other UL subclasses.

Bivalent regions differ between myometrium and UL subclasses. Myometrium bivalent TSSs were deregulated in ULs, seen as activation or repression in UL annotation. This deregulation was reflected in the gene expression levels, validating the functionality of these changes emerging in chromatin annotation level. Activation of bivalent genes was most prominent in MED12 and HMGA2 ULs, possibly linking to the hypermethylation of bivalent regions in these subclasses. We have earlier proposed[4] that bivalent regions and their deregulation play an important role in myometrial gene regulation and UL genesis, and findings of this study further strengthen this hypothesis.

We have previously identified many common[4,27] and rare genetic variants predisposing to ULs[20]. In this study, we identified altogether 149 loci with genome-wide significance. We show uterine tissue to rank

as the most plausible target tissue with gene expression analysis tools FUMA and MAGMA, giving strong evidence for the tissue-specificity of UL predisposing loci. As described before[27] loci involved in genitourinary development and genome stability are highlighted in UL predisposition. The identified loci include genes in both of these categories. *Thymidylate Synthetase* (*TYMS*) and *Poly(ADP-Ribose) Polymerase 1* (*PARP1*) are involved in DNA damage. Hormone receptors *Progesterone Receptor* (*PGR*) and *Estrogen receptor 2* (*ESR2*) were also among the predisposition loci, contributing to the group of genes involved in genitourinary development. We performed FUMA gene-set analysis, which indicated that target genes of our GWAS SNPs are enriched to biological processes involving DNA damage response, telomere organization and cell cycle, further confirming our previous conclusions. Other interesting target genes for UL predisposition include *Mortality Factor 4 Like 2* (*MORF4L2*), which is part of NuA4 histone acetyltransferase complex and works with *YEATS4* and *DMAP1*, members of the SRCAP complex, whose target genes are mutationally inactivated in a subclass of ULs[4]. Among the genes is also *UBE2M*, a gene mutated in a UL subclass defined by defects in Cullin 3-RING E3

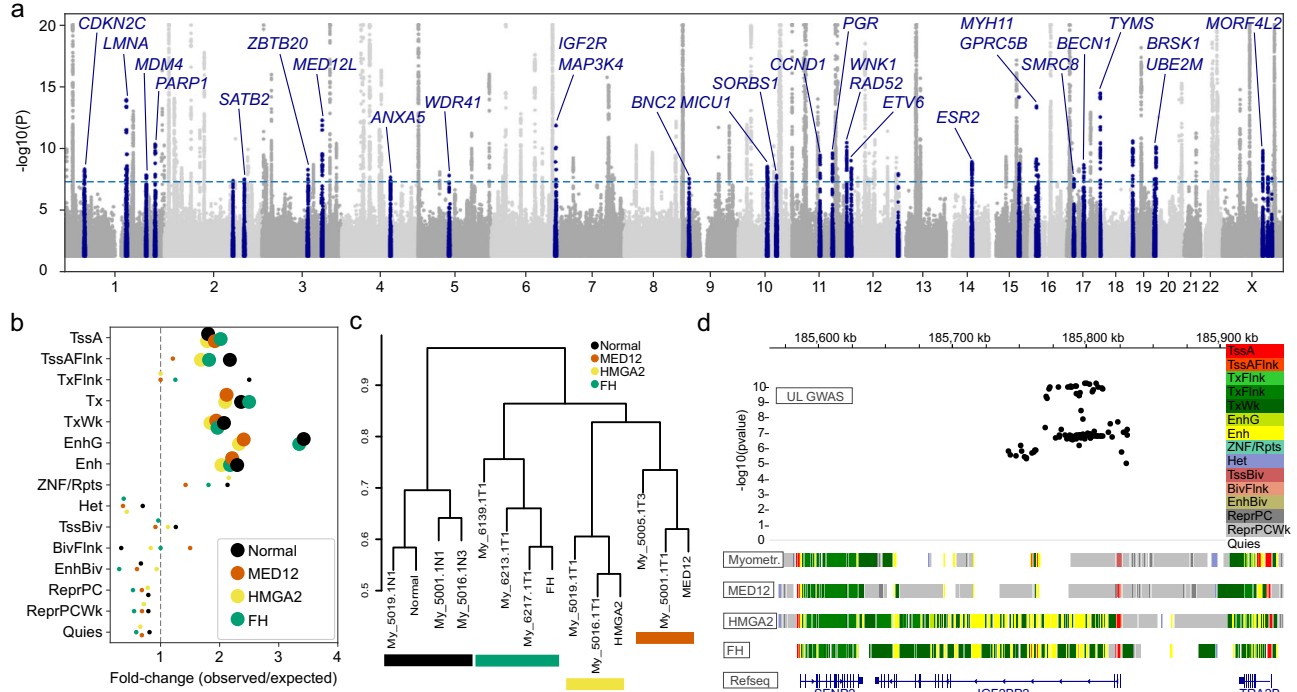

**Fig. 5 | GWAS loci reveal tumor subclass-specific differences in chromatin.**
**a** Manhattan plot of UL associations for 66,075 UL cases and 537,502 female controls (inverse-variance weighted fixed effects meta-analysis P-values in log-scale and truncated to 1e-20); selected loci are highlighted in dark blue color. The dashed line denotes the genome-wide significance threshold (P < 5e-8). **b** UL associated loci were enriched to active chromatin regions in myometrium and in all tumor subclasses (large dots are P < 0.05, small dots P ≥ 0.05; one-sided random resampling test). **c** Hierarchical clustering of the chromatin states at genome-wide significant GWAS loci: the GWAS region chromatin states had more variation between the four

groups (myometrium, MED12, HMGA2, FH) than within these groups.
**d** Representative example of the *IGF2BP2* GWAS locus displaying chromatin state differences among tumor subclasses. TssA Active TSS, TssAFlnk Flanking active TSS, TxFlnk Transcr. at gene 5' and 3', Tx Strong transcription, TxWk Weak transcription, EnhG Genic enhancers, Enh Enhancers, ZNF/Rpts ZNF genes + repeats, Het Heterochromatin, TssBiv Bivalent/poised TSS, BivFlnk Flanking bivalent TSS/Enh, EnhBiv Bivalent enhancer, ReprPC Repressed Polycomb, ReprPCWk Weak repressed Polycomb, Quies Quiescent/low. Source data are provided as a Source Data file.

ligase neddylation[9]. Multiple genes involved in IGF/AKT signaling were also identified; *IGF2R*, *MAP3K4*, *PML*, *PIK3C2B*, *SORBS1*, and *SMCR8*. In addition, one of the identified regions maps downstream of *SATB2*, a developmental gene highly upregulated in ULs[4,17]. With HiChIP data, we could connect this GWAS region to the *SATB2* promoter in ULs but not in myometrium. In addition, *SATB2* downstream region displayed many enhancers and DARs in all UL subclasses, and increased accessibility associated with the risk allele, highlighting the importance of this region in regulation of *SATB2* expression.

Previous efforts have aimed to identify functional consequences of UL GWAS SNPs[28]. In this study, we examined each GWAS locus in light of the 15 state chromatin annotations for myometrium and MED12, HMGA2 and FH ULs. With data from different UL genetic subclasses we were able to identify GWAS regions where the annotation differs between myometrium and even a specific UL genetic subclass, suggesting a possibility of specific regions predisposing to ULs with particular genetic drivers. One of these regions maps to the intron of *IGF2BP2* with enhancers and other active annotations in HMGA2 and FH ULs, whereas in MED12 ULs and myometrium this region is annotated as repressed polycomb. The change in chromatin annotation is also reflected in gene expression levels; *IGF2BP2* is overexpressed in HMGA2 and FH, but not in MED12 ULs compared to myometrium. *HMGA2* overexpression has previously been shown to activate *IGF2BP2* expression in tumorigenesis[16]. The risk allele is slightly more common in patients who developed only *HMGA2* overexpressing ULs, however, this difference remained insignificant. A similar link between a UL risk allele and specific genetic driver has been demonstrated previously for GWAS signal near *MED12* gene, predisposing to *MED12* mutant ULs[27].

We also utilized the power provided by the precise annotations to harvest for candidate predisposition loci of interest that have not reached genome-wide significance in GWAS data. We searched for GWAS loci with suggestive association to ULs and change in the chromatin annotations between myometrium and ULs. Focusing on the more accessible UL enhancers at these regions, we found 60 loci with a more accessible enhancer in each UL subclass. In our previous study[20], one of the identified genes with loss-of-function variants associated with ULs was *BEND3*, in which we now identified a genome-wide suggestive predisposing SNP in the 3'end. *BEND3* is an interesting gene in the context of myometrium and ULs, as it has been previously shown to regulate bivalent genes by preventing their premature activation during differentiation[29]. *BEND3* was also identified as being recruited into chromatin by H2A.Z[30], the histone variant deregulated in ULs[4]. These data point to *BEND3* playing a role in UL genesis, likely through effects on bivalent chromatin.

Taken together, data presented in this study further strengthens the hypothesis that aberrant function of bivalent regions is important for UL genesis. Myometrial tissue is under constant periodic change due to periods of the menstrual cycle and pregnancies, and bivalency is likely to be an important contributor in the associated cyclic regulation. Integration of GWAS and regulatory genomics data from the correct tissue type provided us with deep insight into the regulatory features of the GWAS loci. This approach represents a powerful tool in understanding population-level disease predisposition at the chromatin state level. Mechanistic understanding on how ULs develop is a prerequisite for the development of curative noninvasive management options. Knowledge of the UL regulatory genome appears to be a central part of this puzzle.

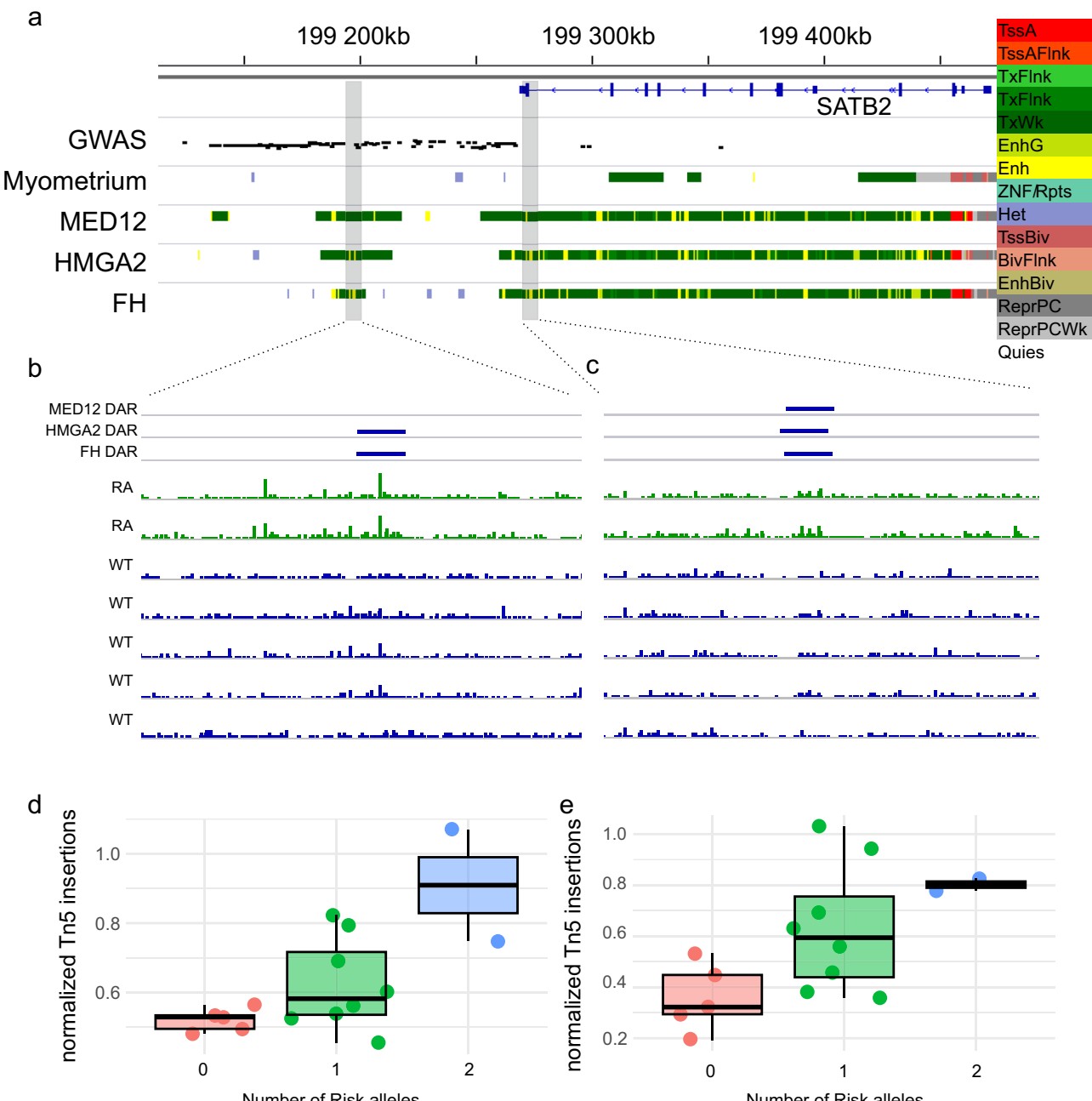

**Fig. 6 | *SATB2* regulatory region is more accessible in GWAS risk allele homozygous myometrium. a** Chromatin annotations and significant GWAS signals at *SATB2* downstream region. The two shaded regions displayed more accessibility in rs7559104 risk allele homozygotes (RA) compared to non-risk allele homozygotes (WT). TssA: Active TSS, TssAFlnk: Flanking active TSS, TxFlnk: Transcr. at gene 5' and 3', Tx: Strong transcription, TxWk: Weak transcription, EnhG: Genic enhancers, Enh: Enhancers, ZNF/Rpts: ZNF genes + repeats, Het: Heterochromatin, TssBiv: Bivalent/poised TSS, BivFlnk: Flanking bivalent TSS/Enh, EnhBiv: Bivalent enhancer, ReprPC: Repressed Polycomb, ReprPCWk: Weak repressed Polycomb, Quies: Quiescent/low. **b** Tn5 shifted fragment ends at chr2:199194872-199199488. **c** Tn5 shifted fragment ends at chr2:199271803-199276398. **d** Chromatin accessibility as normalized fragment end counts (transposase Tn5 insertions per million fragments) for each genotype of *SATB2* risk allele (0: *n* = 5, 1: *n* = 8, 2: *n* = 2 biological

replicates) at chr2:199197134-199197638 (DAR shown in **b**-panel). The risk allele was associated with increased Tn5 insertion counts (linear regression, p-value = 0.004). Centre line, median; box limits, 25% and 75% quartiles; whiskers, 1.5 × interquartile range (IQR) past the quartiles. Association statistics calculated with linear regression. **e** Chromatin accessibility as normalized fragment end counts (transposase Tn5 insertions per million fragments) for each genotype of *SATB2* risk allele (0: *n* = 5, 1: *n* = 8, 2: *n* = 2 biological replicates) at chr2:199273659-199274205 (DAR shown in c-panel). The risk allele was associated with increased Tn5 insertion counts (linear regression, p-value = 0.01). Centre line, median; box limits, 25% and 75% quartiles; whiskers, 1.5 × interquartile range (IQR) past the quartiles. Association statistics calculated with linear regression. Source data are provided as a Source Data file.

## Methods

### Ethics statement

This study was conducted in accordance with the Declaration of Helsinki and approved by the Ministry of Social Affairs and Health and the National Institute for Health and Welfare (53/07/2000, THL/1071/5.05.00/2011, THL/151/5.05.00/2017, THL/1300/5.05.00/2019, THL/1849/14.06.00/2024), and the Ethics Committee of the Hospital District of Helsinki and Uusimaa (133/E8/03, 408/13/03/03/2009, 177/13/

03/03/2016, HUS/2509/2016). The sample set consisted of ULs and myometrium samples collected from 656 hysterectomy patients with a written informed consent.

Patients in the FinnGen cohort have provided written informed consent. Study subjects in FinnGen provided informed consent for biobank research, based on the Finnish Biobank Act. Alternatively, separate research cohorts, collected prior the Finnish Biobank Act came into effect (in September 2013) and start of FinnGen (August 2017), were collected based on study-specific consents and later transferred to the Finnish biobanks after approval by Fimea (Finnish Medicines Agency), the National Supervisory Authority for Welfare and Health. Recruitment protocols followed the biobank protocols approved by Fimea. The Coordinating Ethics Committee of the Hospital District of Helsinki and Uusimaa (HUS) statement number for the FinnGen study is Nr HUS/990/2017. The FinnGen study is approved by Finnish Institute for Health and Welfare (permit numbers: THL/2031/6.02.00/2017, THL/1101/5.05.00/2017, THL/341/6.02.00/2018, THL/2222/6.02.00/2018, THL/283/6.02.00/2019, THL/1721/5.05.00/2019 and THL/1524/5.05.00/2020), Digital and population data service agency (permit numbers: VRK43431/2017-3, VRK/6909/2018-3, VRK/4415/2019-3), the Social Insurance Institution (permit numbers: KELA 58/522/2017, KELA 131/522/2018, KELA 70/522/2019, KELA 98/522/2019, KELA 134/522/2019, KELA 138/522/2019, KELA 2/522/2020, KELA 16/522/2020), Findata permit numbers THL/2364/14.02/2020, THL/4055/14.06.00/2020, THL/3433/14.06.00/2020, THL/4432/14.06/2020, THL/5189/14.06/2020, THL/5894/14.06.00/2020, THL/6619/14.06.00/2020, THL/209/14.06.00/2021, THL/688/14.06.00/2021, THL/1284/14.06.00/2021, THL/1965/14.06.00/2021, THL/5546/14.02.00/2020, THL/2658/14.06.00/2021, THL/4235/14.06.00/2021, Statistics Finland (permit numbers: TK-53-1041-17 and TK/143/07.03.00/2020 (earlier TK-53-90-20) TK/1735/07.03.00/2021, TK/3112/07.03.00/2021) and Finnish Registry for Kidney Diseases permission/extract from the meeting minutes on 4th July 2019. The Biobank Access Decisions for FinnGen samples and data utilized in FinnGen Data Freeze 11 include: THL Biobank BB2017_55, BB2017_111, BB2018_19, BB_2018_34, BB_2018_67, BB2018_71, BB2019_7, BB2019_8, BB2019_26, BB2020_1, BB2021_65, Finnish Red Cross Blood Service Biobank 7.12.2017, Helsinki Biobank HUS/359/2017, HUS/248/2020, HUS/430/2021 §28, §29, HUS/150/2022 §12, §13, §14, §15, §16, §17, §18, §23, §58, §59, HUS/128/2023 §18, Auria Biobank AB17-5154 and amendment #1 (August 17 2020) and amendments BB_2021-0140, BB_2021-0156 (August 26 2021, Feb 2 2022), BB_2021-0169, BB_2021-0179, BB_2021-0161, AB20-5926 and amendment #1 (April 23 2020) and it´s modifications (Sep 22 2021), BB_2022-0262, BB_2022-0256, Biobank Borealis of Northern Finland_2017_1013, 2021_5010, 2021_5010 Amendment, 2021_5018, 2021_5018 Amendment, 2021_5015, 2021_5015 Amendment, 2021_5015 Amendment_2, 2021_5023, 2021_5023 Amendment, 2021_5023 Amendment_2, 2021_5017, 2021_5017 Amendment, 2022_6001, 2022_6001 Amendment, 2022_6006 Amendment, 2022_6006 Amendment, 2022_6006 Amendment_2, BB22-0067, 2022_0262, 2022_0262 Amendment, Biobank of Eastern Finland 1186/2018 and amendment 22§/2020, 53§/2021, 13§/2022, 14§/2022, 15§/2022, 27§/2022, 28§/2022, 29§/2022, 33§/2022, 35§/2022, 36§/2022, 37§/2022, 39§/2022, 7§/2023, 32§/2023, 33§/2023, 34§/2023, 35§/2023, 36§/2023, 37§/2023, 38§/2023, 39§/2023, 40§/2023, 41§/2023, Finnish Clinical Biobank Tampere MH0004 and amendments (21.02.2020 & 06.10.2020), BB2021-0140 8§/2021, 9§/2021, §9/2022, §10/2022, §12/2022, 13§/2022, §20/2022, §21/2022, §22/2022, §23/2022, 28§/2022, 29§/2022, 30§/2022, 31§/2022, 32§/2022, 38§/2022, 40§/2022, 42§/2022, 1§/2023, Central Finland Biobank 1-2017, BB_2021-0161, BB_2021-0169, BB_2021-0179, BB_2021-0170, BB_2022-0256, BB_2022-0262, BB22-0067, Decision allowing to continue data processing until 31st Aug 2024 for projects: BB_2021-0179, BB22-0067,BB_2022-0262, BB_2021-0170, BB_2021-0164, BB_2021-0161, and BB_2021-0169, and Terveystalo Biobank STB 2018001 and amendment 25th Aug 2020, Finnish Hematological Registry and Clinical Biobank decision 18th June 2021, Arctic biobank P0844: ARC_2021_1001.

## ChIP-seq

ChIP-sequencing was performed as described previously[4]. For histone modifications H3K4me1, H3K36me3, H3K4me3, H3K27me3 and H3K9me3 chromatin was crosslinked with 1% formaldehyde. Chromatin was fragmented using micrococcal nuclease (MNase) as described for H3K4me3 and H3K27me3 in ref. 4. Dynabeads protein A (H3K4me3; ThermoFisher Scientific, Cat. No 10002D) or protein G (H3K27me3, H3K36me3, H3K4me1, H3K9me3; ThermoFisher Scientific, Cat. No 10003D) were washed with 0.05% Tween-20 in PBS and incubated with antibody for the ChIP-target and 2µg of spike-in antibody (Active Motif, Cat. No. 61686) 15 min in RT. Used antibodies and lots are listed in Supplementary Table 1. Twenty ng spike-in chromatin (Active Motif, Cat. No. 53083) was added to chromatin from tissue before incubation with the antibody. Sequencing libraries were prepared with TruSeq ChIP Library Preparation Kit and sequenced with HiSeq2500 (H3K4me3, H3K27me3; 20 M 100SE reads) or Novaseq6000 (H3K4me1, H3K36me3, H3K9me3; 20 M 150PE reads).

Raw sequencing reads were processed as described previously[4], except reads were mapped to reference genome GRCh38 (accession: GCA_000001405.15), excluding alt contigs. Briefly, cutadapt version 4.1 in Trim Galore version 0.6.7 with default parameters was used to quality and adapter-trim raw sequencing reads. Trimmed reads were aligned to the GRCh38 reference genome with Bowtie 2 v. 2.5.0. Samtools v. 1.6 was used to filter reads with mapping quality <20 and MACS2 v. 2.2.7.1 was used for peak calling with default parameters, except FDR cutoff for narrowPeak regions was set to 0.01. Each sample included in the study was required to have at least 5% of reads in broadPeaks. The ENCODE blacklist genomic regions[31] were filtered out from the final peaks located at autosomes and the X chromosome.

Locus Overlap Analysis (LOLA) R package v.1.30.0[32] was used to analyze enrichment of H2A.Z narrowPeak regions from pooled myometrium samples on myometrium and UL 15-state segmentations. All regions that had read coverage of at least two in the aligned ChIP–seq data and did not overlap with the ENCODE blacklist genomic regions were considered as a background set.

## Chromatin segmentation and annotation

A 15-state chromatin segmentation was created for myometrium and myoma subclasses MED12, HMGA2, and FH with chromHMM v. 1.24[12] using 2-3 biological replicates for each histone PTM ChIP-seq marker H3K4me1, H3K4me3, H3K36me3, H3K9me3, and H3K27me3 (see details in Supplementary Data 10). In addition, segmentations were created for 2-3 individual samples of each UL subclass, with 1 replicate for each ChIP experiment from the same sample. We used read-level data to create the segmentations, where raw sequencing reads were extended to obtain the mean read length defined for each ChIP-seq experiment. Segmentations were annotated with the Roadmap epigenomics model for 15 chromatin states[3].

Differences in the genome-wide amount of each annotation in base pairs between individual ULs and myometrium samples was tested with a Welch two-sample t-test in R. P-values were adjusted for multiple testing using p-adjust -function and method=holm. Enrichment to myometrium 5-state annotations and RefSeq TSS, TES, Exon, and Gene regions was calculated with chromHMM OverlapEnrichment.

Samples were clustered with the normalized mutual information (NMI) method utilized in package aricode v.1.0.3[33]. For clustering purposes, the whole genome was divided into 200 bp bins, and annotations for each sample were obtained for these bins. All bins with the same annotation in all samples were removed before calculating NMI. NMI was converted to distance and samples were clustered using hclust with method ward.D2.

To compare changes in mapping of chromatin annotation between subclasses the 15 states were combined into seven categories: ActiveTSS (TssA, TssAFlnk), Transcription (TxFlnk, Tx, TxWk), Enhancer (EnhG, Enh), Heterochromatin (ZNF/Rpts, Het), Bivalent (TssBiv, BivFlnk, EnhBiv), RepressedPC (ReprPC, ReprPCWk) and Quiescent (Quies). The set of regions with each of these annotations in myometrium was annotated with UL subclass annotation, and the percentage of each in UL was calculated for the myometrium annotations.

## ATAC-seq

ATAC-seq data was created previously[4]. Sequenced reads were processed as described in ref. [4], except reads were mapped to reference genome GRCh38 (accession: GCA_000001405.15, excluding alt contigs) and fixed-width peaks were generated using MACS2 (v2.1.4) callpeak command with parameters '--shift -100 --extsize 200 --nomodel --call-summits --nolambda --keep-dup all -p 0.01'. More accessible regions (DARs) were calculated for each UL subclass against a pool of 15 myometrium samples, as described previously[4] with DESeq2 v.1.40.2. Enrichment of DARs to myometrium and UL 15-state chromatin segmentations was calculated with Locus Overlap Analysis (LOLA) package[32] v.1.30.0. All open chromatin regions tested for differential accessibility in the UL subclass were used as a background set.

Linear regression was used to calculate the association of Tn5 insertions to the genotype of *SATB2* risk allele in myometrium ATAC-seq samples using the lm-function in R. Tn5 insertions were normalized (Tn5 insertions per million fragments) based on each sample's library size.

We used ChromVAR (v. 1.16) to study transcription factor motif accessibility between FH ULs and myometrium samples at myometrium accessible enhancers hypermethylated in FH. As a motif database we used JASPAR2020[34] human motifs in all versions. Fragments were counted with chromVAR getCounts from bam-files, and deviations with computeDeviations. Differential deviations between FH ULs and myometrium were calculated with chromVARs differentialDeviations.

## HiChIP

HiChIP against H3K27ac was created previously[4], sequenced reads were reprocessed to GRCh38. HiC-Pro v. 3.1.0 was used to identify valid interaction pairs from quality and adapter trimmed reads (Trim Galore v 0.6.7 using Cutadapt v4.1). Bowtie2 v.2.4.4 was used to map reads to the reference genome GRCh38 (accession: GCA_000001405.15) with HiC-Pro default parameters and minimum mapping quality 10. Reads were paired and assigned to restriction fragment GATCGATC. Self-circle and dangling end fragments were discarded, and valid interaction pairs were detected. FitHiChIP v11.0 was used to call significant interactions from all valid pairs identified by HiC-Pro. Interactions were called with bin size 5 kb, with lower and upper distance threshold of loops 10 kb and 10 Mb, respectively, and with FDR threshold 0.01 for significant loops. Coverage bias correction and peak to all (L) background were used. H3K27ac ChIP-seq narrow peaks were used to identify interactions related to binding of H3K27ac. Differential analysis of FitHiChIP significant loops between pooled ULs and pooled myometrium samples was run with FitHiChIP using EdgeR v.3.32.1 and default parameters, including --FoldChangeThr 2 and --DiffFDRThr 0.05.

## Determination of myometrium bivalent genes

The closest transcription start site of UL/myometrium expressed genes was annotated for each region annotated as TssBiv in myometrium. Genes were classified as bivalent in myometrium if the distance of TssBiv annotation from the transcription start site was less than or equal to 1000 bp. Pheatmap v.1.0.12 was used to cluster myometrium and UL samples based on expression of the myometrium bivalent genes with method ward.D2.

Each myometrium bivalent TSS region (TSS + -1000bp) was annotated in ULs, and regions not overlapping UL bivalent annotations (TssBiv, BivFlnk and EnhBiv) were determined as activated (>50% of TSS region annotated with TssA or TssAFlnk), repressed (>50% of TSS region annotated with ReprPC or ReprPCWk) or other in UL subclasses. Expression of UL activated and repressed genes was plotted with ggplot2 v.3.4.3.

## Genome-wide association study

Inherited uterine leiomyoma risk was examined by implementing a meta-analysis combining three genome-wide association studies (GWAS): Biobank Japan (BBJ), http://jenger.riken.jp/en/, accessed on Sept 2, 2020), UK Biobank (UKB, https://www.ukbiobank.ac.uk/, Application Number 80756, accessed on Jun 11, 2021) and FINNGEN (https://finngen.fi/, release 12, accessed on Dec 15, 2023). The BBJ cohort had a total of 5,954 UL cases and 95,010 female controls. The UK Biobank cohort of white British women was analyzed for self-reported ULs and ICD10/9 codes (see Berta et al. [4] for phenotype definition, population stratification and genotype quality-control steps), resulting in a total of 18,014 UL cases and 202,535 female controls. The FINNGEN cohort had a total of 42,107 UL cases and 239,957 female controls. All cohort-wise summary statistics were precomputed with mixed model logistic regression (SAIGE); details are available at the sources listed above. The BBJ and UKB data were lifted over to GRCh38/hg38 coordinates (Picard LiftoverVcf). An inverse-variance weighted fixed effects meta-analysis (R package 'meta' v4.8-4) was applied to 12.8 million imputed SNPs, of which 6.3 million SNPs were available from all three cohorts and 6.5 million SNPs were available only from FINNGEN and UKB. Due to the overwhelming number of UL cases in FINNGEN, we report the best available association from either the meta-analysis or FINNGEN alone. Supplementary Data 5 gives a short-list of the most significant associations in an 1Mbp window that passed $P < 5\times10^{-8}$, including the summary statistics for each of the three cohorts. Previously published UL predisposition loci were collected from Berta et al. [4], Sliz et al. [35] and GWAS Catalog (v1.0 accessed on Sept 9, 2020). Enrichment to the chromatin annotation was measured by overlap between the genome-wide significant GWAS SNPs ($P < 5\times10^{-8}$) and each annotation; fold-change and significance of the enrichment were estimated by comparing against 10,000 randomly distributed resamples of the annotation.

Functional mapping and annotation of GWAS SNPs were performed using FUMA[18] (v1.5.2, accessed on Dec 3, 2024), including MAGMA[19] (v1.08) tissue expression analysis. Genome-wide summary statistics were first lifted-over to hg19 coordinates (Picard LiftoverVcf), followed by FUMA and MAGMA analyses using their default settings: protein-coding genes from Ensembl v102, positional mapping with a 10Kbp flank, and GTEx v8.

GWAS SNPs (Supplementary Data 5) were tested for eQTLs for all genes within 500Kbp. Gene expression data was available from 152 myometria (unique patients) and 231 ULs (at most one tumor per patient included in the analysis). The eQTL association tests (DESeq2 v1.46.0, two-sided Wald test) included all GWAS SNPs with a minimum of 10 alternative allele carriers and were adjusted for genetic ancestry (first four principal components), patients' age at hysterectomy and UL subclass. Genes with low expression (base mean <10) were excluded.

## GWAS region annotation

We created GWAS regions from genome-wide significant SNPs ($P < 5\times10^{-8}$) by the SNP with the smallest p-value at each region +2000bp flanks. Six regions had two SNPs with the same smallest p-value, and both of these SNPs were taken for further analyses.

We annotated the GWAS regions with 15-state chromatin annotation for myometrium and each UL subclass. The regions were divided into five categories based on myometrium annotations: Transcription (>=60% of region annotated as Tx and/or TxWk; 55 regions), Transcription and/or enhancer (38 regions), Repressed Polycomb (>=50% ReprPC and/or ReprPCWk; 20 regions), Quiescent (>=80% Quies or Het; 27 regions) and chromosome X (10 regions).

We used normalized mutual information (NMI) method in aricode v.1.0.3[33] to cluster chromatin segmentations based on annotations at GWAS regions (most significant SNP +2000bp flanks). Chromatin annotations in 200 bp bins overlapping these regions were used to calculate NMI, and the result was converted into distance and clustered using hclust with method ward.D2.

**Identification of more accessible UL enhancers and mapping of target genes**

To identify more accessible UL enhancers, we combined all genic enhancers (EnhG) and enhancers (Enh) in each subclass. UL enhancers overlapping with myometrium enhancers were removed, and enhancer accessibility was calculated from ATAC-seq fragment end counts, as described previously[4] with DESeq2 v.1.40.2. Enhancers with more accessibility (log2FC > 0, padj<0.05) were defined as more accessible UL enhancers.

We annotated the closest TSS of a protein-coding gene expressed in myometrium and/or ULs with bedtools v.2.31.0 closest -function in bedtoolsr wrapper. GRCh38 TSS was obtained from biomaRt v.2.58.2 ensembl v. GRCh38.p14.

ENCODE rE2G thresholded links were downloaded for 352 cell/tissue types. Links were filtered for protein-coding myometrium/UL expressed genes, and links with regulatory region overlapping more accessible enhancers were searched with bedtools v.2.31.0 intersect-function in bedtoolsr wrapper.

HiChIP differential interactions with 5k bin size between pooled myometrium ($n = 5$) and UL ($n = 5$) samples were used to map target genes for more accessible UL enhancers. All differential interactions with a more accessible UL enhancer were determined, and all genes with TSS in the other end bin of the interaction were taken into further analyses. UpsetR v.1.4.0, pheatmap v.1.0.12 and ggplot2 v.3.4.3 were used for visualization.

**CpG methylation enrichment**

Differentially methylated loci (DMLs)[4] were analyzed for enrichment to the 15 chromatin states for ULs and myometrium with Locus Overlap Analysis (LOLA) R package[32] v.1.30.0. Enrichment was calculated separately for hypo- and hypermethylated loci. All CpG-containing regions that were tested for DMLs were included in the background set.

**Reporting summary**

Further information on research design is available in the Nature Portfolio Reporting Summary linked to this article.

## Data availability

The ATAC-seq, RNA-seq, CpG methylation, ChIP-seq against H2A.Z, and HiChIP publicly available data used in this study are available in the European Genome–phenome Archive database under accession code EGAS00001004499[4]. The 15-state chromatin segmentations and ChIP-seq data generated in this study have been deposited in the Federated European Genome-phenome Archive database under accession code EGAD50000001443. Due to the sensitive nature of genetic data access is restricted, and can be obtained in FEGA. Data access requests will be evaluated by the Data Access Committee in accordance with the Finnish legislation and the European General Data Protection Regulation (GDPR), and can be granted to non-commercial academic research on

neoplasia. Data requests will be processed in four weeks. Data access period is six months, after which an extension time can be applied. The 15-state chromatin segmentations for myometrium and the UL subclasses can be accessed also through Zenodo [https://doi.org/10.5281/zenodo.13373492]. Differential gene expression data can be accessed in Berta et al.[4] Supplementary Table 21. Used ENCODE rE2G accession numbers are listed in Supplementary Data 11. The previously published GWAS data used in this study are publicly available in Berta et al.[4] Supplementary Table 25; in Sliz et al.[35] Table 1; and in GWAS Catalog. (https://www.ebi.ac.uk/gwas/downloads, accessed on Sept 9, 2020). Source data are provided with this paper. The remaining data are available within the Article, Supplementary Information or Source Data file. Source data are provided with this paper.

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

## Acknowledgements

We thank Alison London, Sini Marttinen, Heikki Metsola, Marjo Rajalaakso, Janne Ravantti, Sirpa Soisalo, Iina Vuoristo and Inga-Lill Åberg for technical support. The authors acknowledge the computational resources provided by the ELIXIR node, hosted at the CSC-IT Center for Science, Finland. This research was conducted with the UK Biobank Resource under application number 80756. We want to acknowledge the participants and investigators of the Biobank Japan Project for the 'uterine fibroids' GWAS summary statistics. This study was funded by the Research Council of Finland (The Finnish Centre of Excellence in Tumor Genetics Research Program 2018–2025 (No. 352814, L.A.A) and Academy Professor grants (No. 319083 and 320149, L.A.A)), Jane and Aatos Erkko Foundation (No 220001, L.A.A), Sigrid Jusélius Foundation (No. 230002, L.A.A), Orion Research Foundation sr (M.R) and Cancer Foundation Finland sr (M.R). We want to acknowledge the participants and investigators of the FinnGen study. The FinnGen project is funded by two grants from Business Finland (HUS 4685/31/2016 and UH 4386/31/2016) and the following industry partners: AbbVie Inc., AstraZeneca UK Ltd, Biogen MA Inc., Bristol Myers Squibb (and Celgene Corporation & Celgene International II Sàrl), Genentech Inc., Merck Sharp & Dohme LCC, Pfizer Inc., GlaxoSmithKline Intellectual Property Development Ltd., Sanofi US Services Inc., Maze Therapeutics Inc., Janssen Biotech Inc, Novartis AG, and Boehringer Ingelheim International GmbH. Following biobanks are acknowledged for delivering biobank samples to FinnGen: Auria Biobank (www.auria.fi/biopankki), THL Biobank (www.thl.fi/biobank), Helsinki Biobank (www.helsinginbiopankki.fi), Biobank Borealis of Northern Finland (https://www.ppshp.fi/Tutkimus-ja-opetus/Biopankki/Pages/Biobank-Borealis-briefly-in-English.aspx), Finnish Clinical Biobank Tampere (www.tays.fi/en-US/Research_and_development/Finnish_Clinical_Biobank_Tampere), Biobank of Eastern Finland (www.ita-suomenbiopankki.fi/en), Central Finland Biobank (www.ksshp.fi/fi-FI/Potilaalle/Biopankki), Finnish Red Cross Blood Service Biobank (www.veripalvelu.fi/verenluovutus/biopankkitoiminta), Terveystalo Biobank (www.terveystalo.com/fi/Yritystietoa/Terveystalo-Biopankki/Biopankki/) and Arctic Biobank (https://www.oulu.fi/en/university/faculties-and-units/faculty-medicine/northern-finland-birth-cohorts-and-arctic-biobank). All Finnish Biobanks are members of BBMRI.fi infrastructure (https://www.bbmri-eric.eu/national-nodes/finland/). Finnish Biobank Cooperative -FINBB (https://finbb.fi/) is the coordinator of BBMRI-ERIC operations in Finland. The Finnish biobank data can be accessed through the Fingenious® services (https://site.fingenious.fi/en/) managed by FINBB.

## Author contributions

M.R., E.K., D.G.B., N.V., and L.A.A conceptualized the study. M.R., M.J., and D.G.B. performed ChIP-seq, ATAC-seq and HiChIP experiments. M.R., E.K., and N.V. analyzed ChIP-seq data and created chromatin segmentations. M.R. and E.K. analyzed ATAC-seq and HiChIP data. E.K. and A.T. analyzed methylation data. M.R. & N.V. analyzed RNA-seq data. FinnGen and N.V. conducted genome-wide association studies. E.S., R.B., O.H., and A.P. collected the patient samples. A.K. supervised UL subtyping. M.R., N.V., E.K., and L.A.A. wrote the manuscript. All authors read and approved the manuscript.

## Competing interests

The authors declare no competing interests.

## Additional information

## FinnGen

**Niko Välimäki** [1,2], **Oskari Heikinheimo** [4] **& Lauri A. Aaltonen** [1,2,5] ✉

A full list of members and their affiliations appears in the Supplementary Information.

