## [Transparent Peer Review file · Nature Communications]

Chromatin state origins of uterine leiomyoma

Corresponding Author: Professor Lauri Aaltonen

Version 0:

Reviewer comments:

Reviewer #1

(Remarks to the Author)

Uterine leiomyomas (UL) also known as uterine fibroids is a severe disease with benign tumors that affect 65-70 % of all women in their lifetime. The pathological effect is manifested by unusual deposition of extracellular matrix protein rendering uterine tissues highly fibrotic, and stiff. Several genetic alterations that are possibly mutually exclusive, (MED12 mutation, biallelic inactivation of fumarate hydratase, overexpression of HMGA2 and deficient H2AZ loading by mutations in SRCAP complex and deletions of the COL4A5/6 locus. Previous work from this group and others using human samples provided an extensive mapping of the leiomyoma genome and epigenome. For example, in previous work this group and other groups have used epigenomic features to distinguish between normal and leiomyoma samples and in some cases leiomyoma subtypes.

In this work, the authors included additional epigenomic marks, and RNA sequencing and ATAC sequencing and further refined chromatin state annotation of myometrium and 3 UL subclasses namely, MED12 mutants, HMGA2 overexpressors and FH mutants. Using extensive bioinformatic analysis the authors classified 15 chromatin states (encompassing 7 broader categories) that the authors the current work confirms and also extends previous genome organization data.

The major strengths of the work is that it now adds an extensive set of new features of chromatin states (active enhancers, bivalent promoters, heterochromatin, active TSS and etc.) of leiomyoma subtypes and “normal” myometrium. In that sense this current work extends previous observations and refines chromatin states: This extensive data sets and analysis will be very helpful to the scientific community studying UL.

While this is nice and good, it does not reveal significant novelty and thus overall impact in the field and beyond and its appropriateness for being published in a broader (not specialized) journal like Nat. Comm.

As impressive as the data analysis, it remains descriptive in nature. The biggest limitations of this work that it does not provide any functional validations for some of the key observations. Much of the new GWAS association need to be experimentally validated (at the minimum some selected loci covering SATB2, SER1,2, PGR, IGF2BP2 etc.). Recently published data on GWAS analysis of leiomyoma and engineered mutant cells may provide a platform for genotype specific mechanistic and functional validations. Otherwise the current work confirms and validates previous work.

How does K27 Hi ChIP compares with promoter capture HiC data sets?

Are 3 normal Myo samples represent UL adjacent “non diseased tissue samples?

In Fig 2 addition of H2AZ myo binding does not add significantly to the study.

How do the authors explain the results that In FH samples enhancers are hypermethylated but has open chromatin structure in enhancers when compared to similar open chromatin structures in HMGA2 and MED12 samples where enhancer are hypomethylated.

Defining the bivalent regions are very interesting and informative. However in Fig 3 why FH samples don't show any changes active TSS when compared to MED12 and HMGA2 samples?

Figs 4, 5: GWAS SNPs on SATB2/BEND3 and possibly on other UL associated genes in chromosome domain interactions are interesting and correlative. Mechanistic experiments need to be included to validate some of the GWAS observation to

chromatin states and gene expression.

Observations described in Fig 2B linking FH deficiency, TET2 regulation and DNA methylation status are very nice.

In summary the manuscript is descriptive in nature, adds important and new information in the area of UL, but lacks mechanistic and functional studies which will be important requirements for the appropriateness of this work for Nat. Comm. Otherwise as presented, it is more appropriate for a specialized journal.

Reviewer #2

(Remarks to the Author)

Raisanen et al. analysed the state of chromatin in detail based on multi-layered omics data for uterine leiomyoma (UL) and myometria, and identified differences in chromatin state among three UL subclasses and myometrium.

In addition, they conducted trans-ancestry GWAS meta-analysis consisting of 66075 UL cases and 537502 controls, and identified 149 significant loci including 35 novel loci. They found that GWAS loci were enriched at active chromatin, especially at enhancers, and harbored tumor-specific, and driver-specific chromatin state changes.

The authors are using a variety of the latest analytical methods to evaluate the chromatin state of UL in detail, and is also utilize their results for post-GWAS analysis. Their research is considered to be a very important achievement that will lead to the understanding of the mechanisms of disease onset. I have a few comments as follows.

1. The study design and data used in paper is unclear:

It is necessary to clearly state in the text and in Fig. 1a whether the data used in this study is from a previous paper (Berta et al. Nature 2021) or newly analysed data. Please clearly state the differences between existing data sets and new data is important for the reliability and transparency of the study.

2. Sample collection

In Fig 1a, Finland myoma study was shown to include 656 Patients and 2061 UL (tissues?). The authors need to clarify whether multiple UL (uterine leiomyoma) tissues were taken from one patient or not. In addition, the origin of control myometria tissues should be described (surrounding normal tissues surgically resected for the operation of UL?).

3. SNPs with significant associations in the GWAS meta-analysis and eQTL effects:

The authors analyzed 266 UL and 153 myometria RNA seq-analysis. It is very important to evaluate whether the SNPs found in the GWAS meta-analysis show eQTL effects. It is also important to check whether the results of these SNPs are consistent with the results of chromatin annotation. This will clarify the relationship between genetic variation and chromatin structure.

4. Tissue specificity of GWAS loci

It is useful to use analysis tools such as MAGMA and FUMA to verify tissue-specific genetic associations. This may reveal genetic factors that are specific to a particular tissue (e.g.uterus). This will provide clues to the tissue-specific mechanisms of the disease.

Version 1:

Reviewer comments:

Reviewer #1

(Remarks to the Author)

The authors have sincerely addressed questions/suggestions from both reviewers: These include revision of texts and abstracts, inclusion of statements for clarity, and new figures. The manuscript should be acceptable for publication.

Reviewer #2

(Remarks to the Author)

The authors fully addressed this reviewer's comments in the Revised paper. We believe that this paper is appropriate for publication in nature communications.

POINT BY POINT RESPONSE TO REVIEWERS' COMMENTS

Reviewer #1, expertise in multi-omics for uterine leiomyomas (Remarks to the Author):

Uterine leiomyomas (UL) also known as uterine fibroids is a severe disease with benign tumors that affect 65-70 % of all women in their lifetime. The pathological effect is manifested by unusual deposition of extracellular matrix protein rendering uterine tissues highly fibrotic, and stiff. Several genetic alterations that are possibly mutually exclusive, (MED12 mutation, biallelic inactivation of fumarate hydratase, overexpression of HMGA2 and deficient H2AZ loading by mutations in SRCAP complex and deletions of the COL4A5/6 locus. Previous work from this group and others using human samples provided an extensive mapping of the leiomyoma genome and epigenome. For example, in previous work this group and other groups have used epigenomic features to distinguish between normal and leiomyoma samples and in some cases leiomyoma subtypes.

In this work, the authors included additional epigenomic marks, and RNA sequencing and ATAC sequencing and further refined chromatin state annotation of myometrium and 3 UL subclasses namely, MED12 mutants, HMGA2 overexpressors and FH mutants. Using extensive bioinformatic analysis the authors classified 15 chromatin states (encompassing 7 broader categories) that the authors the current work confirms and also extends previous genome organization data.

The major strengths of the work is that it now adds an extensive set of new features of chromatin states (active enhancers, bivalent promoters, heterochromatin, active TSS and etc.) of leiomyoma subtypes and “normal” myometrium. In that sense this current work extends previous observations and refines chromatin states: This extensive data sets and analysis will be very helpful to the scientific community studying UL.

We appreciate the reviewer acknowledging the strengths in our study and the value of the extensive data sets we have created for the scientific community. Indeed, the data created in this study will enable others to leverage the chromatin state annotations, together with other extensive data sets we have published previously (Berta et al. 2021), to better understand the regulatory genome functions in myometrium and ULs with various genetic drivers. Given that myometrium tissue is not analyzed in large public data sets (Roadmap epigenomics, ENCODE), the data created in this study presents a valuable resource to study this specific tissue type with vast impact on women's health.

While this is nice and good, it does not reveal significant novelty and thus overall impact in the field and beyond and its appropriateness for being published in a broader (not specialized) journal like Nat. Comm.

This is a ground-breaking work where a 15-state chromatin annotation was first created from the ideal primary tissue type, and subsequently integrated with previously known and novel GWAS signals. This, and the added layer of normal-tumor characterization of changing

chromatin states, present to our knowledge a unique setting to study the germline predisposition to neoplasia, and pave the way to next level understanding of disease predisposition in general, and tumor predisposition in particular.

The large data sets created in this study and previously (Berta et al. 2021) will enable researchers to study the epigenomes and gene expression of distinct UL subclasses and myometrium. Previous UL studies on the epigenome are mostly focused on MED12 ULs (Moyo et al. 2020), and our data broadens the view to other genetic subclasses highlighting differences between them. We have mapped the bivalent genes in myometrium, and show gene expression changes in ULs when bivalency is lost in tumorigenesis shedding light into bivalency as an important contributor in UL genesis.

We have also conducted the largest-to-date UL GWAS meta-analysis combining three cohorts (UKB, BBJ and FinnGen). We report 35 novel loci contributing to UL predisposition, and show that the signal is enriched at enhancers in our chromatin annotation, highlighting the importance of using the correct tissue type in these analyses. In addition, we utilize the genome annotation differences between ULs and myometrium to identify and characterize novel loci with weaker GWAS signals. This approach gives us more confidence on the importance of these regions in UL genesis, while GWAS signal alone would not reach genome-wide significance leaving these loci undetected.

As impressive as the data analysis, it remains descriptive in nature. The biggest limitations of this work that it does not provide any functional validations for some of the key observations. Much of the new GWAS association need to be experimentally validated (at the minimum some selected loci covering SATB2, SER1,2, PGR, IGFBP2 etc.). Recently published data on GWAS analysis of leiomyoma and engineered mutant cells may provide a platform for genotype specific mechanistic and functional validations. Otherwise the current work confirms and validates previous work.

To describe this ground-breaking work as descriptive appears harsh.

In this work, we report 149 UL predisposition loci, of which 35 are novel associations to the disease. Performing cell model perturbations for any significant number of these to functionally validate the effect of individual variants on gene expression would require a massive effort with limited added value for the current work.

We appreciate the request to provide further validation for the results, and believe that the data from the actual tissues in question are superior to that derived from model systems where manipulated cells float in artificial media. We have previously created ATAC-seq data for 15 myometrium samples (Berta et al. 2021), which allowed us to directly measure the effect of the GWAS SNP on chromatin accessibility. There were multiple loci that allowed us to perform a meaningful comparison of homozygous wild-types versus homozygous risk allele carriers. We focused on *SATB2* - a novel predisposition locus identified in the study - to demonstrate how chromatin accessibility varies between the genotypes at inherited UL risk

loci already in myometrium tissue at regions where more accessibility is gained during tumorigenesis. We compared two individuals homozygous for the risk allele (RA) against five individuals homozygous for the non-risk allele (WT), and show how two regions that are more accessible in ULs are already more accessible in RA myometrium compared to WT myometrium (see **Fig. 6**). The chromatin accessibility is significantly associated with genotype of the risk allele. Chromatin accessibility of heterozygotes is in between the two groups of homozygotes (Fig. 6d-e, see also revised manuscript).

This provides insight into differences on the chromatin architecture present already in myometrium tissue, being dependent on the individual's genotype for the UL predisposing SNPs. We assessed chromatin accessibility also at other key informative GWAS loci *ESR2* and *PGR*. These did not display accessibility differences based on UL risk genotype.

In this study, we conducted the largest UL GWAS analysis to date, identifying new plausible target genes. We performed “Generalized Gene-Set Analysis of GWAS Data” (MAGMA) and “Functional Mapping and Annotation of GWAS” (FUMA) analyses aiming to validate functionality of plausible target genes and tissue specificity of our GWAS results. We have added these results in text (lines 367-382). These results strengthened our claims on two fronts: First, the FUMA gene-set analysis showed strong enrichment for biological processes involved in DNA damage and telomere organization, which validates our original conclusions about the roles these GWAS loci have in UL tumorigenesis (Välimäki et al. 2018). Second, the FUMA differential expression analysis revealed the strongest enrichment for uterine tissue among the 54 different tissue types examined, supporting the notion that our GWAS signals arise from loci that are specific to this tissue type.

How does K27 Hi ChIP compares with promoter capture HiC data sets?

Promoter-capture HiC -data represents interactions enriched for established promoter sequences. These data have been published for myometrium (n=5) and MED12 ULs (n=5) (Moyo et al. 2020). On the other hand, HiChIP enriches interactions associated with binding

of a target protein, in our case histone PTM H3K27ac, which marks active enhancers and promoters (Berta et al. 2021). Thus, H3K27ac-HiChIP represents the active enhancer-promoter and enhancer-enhancer interactions, while promoter-capture HiC enriches all promoter sequences from the used panel.

We analyzed two MED12-UL samples (LEIO_PT916, LEIO_PT886) from the promoter capture HiC -dataset with the same HiC-Pro configurations as we have used with the HiChIP data. We chose these specific samples to represent promoter-capture HiC datasets, as their ethnicity matched our sample set. We compared our HiChIP data from one MED12 UL against promoter-capture HiC data with HiCPlotter (Akdemir and Chin 2015) using 40kb resolution, and show how the interactomes correlate between these datasets from individual MED12 ULs.

Below we show a region in chromosome 2 (chr2:195960000-200000000, 40kb resolution, contains *SATB2*), showing matching interaction nodes in HiChIP and promoter-capture HiC.

Similarly, below we show a region in chromosome 6 (chr6:147960000-154000000, 40kb resolution, contains *ESR1*). Here, interaction nodes are also reproduced with both methods, giving confidence on the data.

HiChIP seems to enrich intra-TAD interactions compared to promoter-capture HiC, while longer-range contacts seem to be stronger in promoter-capture HiC. Not all interaction domains identified with HiChIP are recovered with promoter-capture HiC, possibly due to gene density of these regions or enhancer-enhancer interactions. As promoter-capture HiC enriches contacts to promoter sequences, interaction domains without promoters remain undiscovered.

We looked specifically at the enhancer region downstream of *SATB2* that we identified in the present study, and interactions from this locus to the gene promoters. With 40kb resolution, we see many contacts at the enhancer location interacting with *SATB2* promoter in both data. This gives confidence in our findings on the downstream enhancer with novel predisposition signal and *SATB2* as the target. Two blue boxes below each heatmap show the location of the enhancer (smaller blue box) and the *SATB2* gene (larger blue box).

Both presented methods aiming to identify 3D interactions have their limitations, but these independent data created from ULs and myometrium having similar regulatory contacts and interaction nodes in the key UL-GWAS loci gives confidence on the biological relevance of these regions in the gene regulation in ULs.

Are 3 normal Myo samples represent UL adjacent “non diseased tissue samples?”

The presented myometrium samples are myometrium tissue collected from UL patients at hysterectomy, from sites not adjacent to tumors. We have clarified this statement in the text (line 90-92).

In Fig 2 addition of H2AZ myo binding does not add significantly to the study.

We have revised the figure as the reviewer suggested and removed the corresponding figure panel from Fig. 2.

How do the authors explain the results that In FH samples enhancers are hypermethylated but has open chromatin structure in enhancers when compared to similar open chromatin structures in HMGA2 and MED12 samples where enhancer are hypomethylated.

We thank the reviewer for this insightful question. In FH ULs, both more accessible regions (DARs) and hypermethylated CpGs indeed are enriched at enhancers, however, these are not the same set of enhancers, as shown in the following analysis:

We extracted all FH enhancers (7_Enh in the 15-state chromatin model) overlapping DARs in FH ULs, and all FH enhancers overlapping hypermethylated CpGs in FH. Only 19 enhancers that had overlapping DAR harbored also hypermethylated CpG in FH ULs. Thus, while both of these modifications were enriched at enhancer regions, they appeared mutually exclusive. We have added this notion to the results (line 178-180, Supplementary Figure 1b).

This result is in line with our notion of reduced accessibility of ESR1 and 2 motifs at hypermethylated enhancers in FH ULs compared to myometrium. Hypermethylation of enhancers with ESR1 and 2 motifs in FH ULs leads to reduced accessibility.

Defining the bivalent regions are very interesting and informative. However in Fig 3 why FH samples don't show any changes active TSS when compared to MED12 and HMGA2 samples?

We appreciate the reviewer finding our results interesting. Indeed, in FH ULs change in chromatin annotation from bivalent to active TSS does not create as strong an effect on gene upregulation as is clearly seen in MED12 and HMGA2 ULs in Fig. 3d. The mean log2 fold change of genes with activated TSS in FH is 0.18 (MED12=0.36 and HMGA2=0.33). We have clarified the mean expression change in activated genes in text (line 217-220).

In MED12 and HMGA2 ULs, hypermethylation is enriched at bivalent regions, which is not the case in FH ULs (Fig. 2b). Perhaps the alterations on bivalent chromatin are more important in the genesis of HMGA2 and MED12 ULs than in FH ULs, where metabolic changes drive UL genesis. Thus, we see more data layers pointing toward deregulation of bivalency in MED12 and HMGA2 ULs, whereas in FH ULs the effect is not as striking. Nevertheless, the effect of gene downregulation at promoters with bivalent annotation changing to Repressed polycomb can be seen in all three UL subclasses, pointing to regulatory functions of the chromatin annotation change at myometrial bivalent regions in all UL subclasses.

Figs 4, 5: GWAS SNPs on SATB2/BEND3 and possibly on other UL associated genes in chromosome domain interactions are interesting and correlative. Mechanistic experiments need to be included to validate some of the GWAS observation to chromatin states and gene expression.

We are not entirely sure what kind of experiments the Reviewer is referring to here. In general, we refer to the reply provided above to a similar question regarding functional analyses.

We have added results of the genotype effect on the chromatin accessibility at *SATB2* -locus (Fig. 4e) as described above (rebuttal page 2-3).

BEND3 is not shown in figures 4 or 5. Perhaps the Reviewer means *IGF2BP2* shown in Fig. 5d? Fig. 5d presents an example of overlapping GWAS signal and chromatin annotation differences specific for UL subclasses at *IGF2BP2* -locus. As expected, active chromatin annotations in HMGA2 and FH ULs do indeed connect to higher gene expression of *IGF2BP2* in RNA sequencing (please see the enclosed figure). In contrast, in MED12 ULs without active chromatin annotations in the locus, the gene is not overexpressed. The overlapping GWAS signal in this locus highlights its importance in UL genesis, possibly in a subclass-specific manner.

Observations described in Fig 2B linking FH deficiency, TET2 regulation and DNA methylation status are very nice.

We thank the reviewer for the comment.

In summary the manuscript is descriptive in nature, adds important and new information in the area of UL, but lacks mechanistic and functional studies which will be important requirements for the appropriateness of this work for Nat. Comm. Otherwise as presented, it is more appropriate for a specialized journal.

Please see our responses above, where we respectfully disagree with the overall assessment. We thank the reviewer for the comments and suggestions for improving this manuscript.

Reviewer #2, expertise in uterine leiomyomas genetics and GWAS (Remarks to the Author):

Raisanen et al. analysed the state of chromatin in detail based on multi-layered omics data for uterine leiomyoma (UL) and myometria, and identified differences in chromatin state among three UL subclasses and myometrium.

In addition, they conducted trans-ancestry GWAS meta-analysis consisting of 66075 UL cases and 537502 controls, and identified 149 significant loci including 35 novel loci. They found that GWAS loci were enriched at active chromatin, especially at enhancers, and harbored tumor-specific, and driver-specific chromatin state changes.

The authors are using a variety of the latest analytical methods to evaluate the chromatin state of UL in detail, and is also utilize their results for post-GWAS analysis. Their research is considered to be a very important achievement that will lead to the understanding of the mechanisms of disease onset. I have a few comments as follows.

We thank the reviewer for the encouraging words and acknowledging the importance of the work.

1. The study design and data used in paper is unclear:

It is necessary to clearly state in the text and in Fig. 1a whether the data used in this study is from a previous paper (Berta et al. Nature 2021) or newly analysed data. Please clearly state the differences between existing data sets and new data is important for the reliability and transparency of the study.

We thank the reviewer for the comment, and agree that this aspect needs more clarity. We have modified Fig1. a-panel to better represent which data is from previous publication and which is new to this study. This has also been clarified in the text (line 79).

2. Sample collection

In Fig 1a, Finland myoma study was shown to include 656 Patients and 2061 UL (tissues?). The authors need to clarify whether multiple UL (uterine leiomyoma) tissues were taken from one patient or not. In addition, the origin of control myometria tissues should be described (surrounding normal tissues surgically resected for the operation of UL?).

656 patients from the Finland myoma study contributed to this study. We have collected 2061 ULs from these patients for genotyping, however, in downstream analyses only one UL per patient has been utilized. This has now been clarified in the figure legend of Fig. 1a.

Myometrium tissue used in the analyses is collected from UL patients after hysterectomy, from sites distant from the tumor. We have clarified this statement in the text (line 90-92).

3. SNPs with significant associations in the GWAS meta-analysis and eQTL effects:

The authors analyzed 266 UL and 153 myometria RNA seq-analysis. It is very important to evaluate whether the SNPs found in the GWAS meta-analysis show eQTL effects. It is also important to check whether the results of these SNPs are consistent with the results of chromatin annotation. This will clarify the relationship between genetic variation and chromatin structure.

We agree with the reviewer that a closer analysis of eQTL effects is important for clarifying the relationships between the different layers of data. To address this, we have now included

three new analyses in the manuscript that integrate chromatin accessibility, gene expression and GWAS data. First, we implemented the eQTL analysis requested by the reviewer; the resulting UL and myometria eQTLs are enumerated in Supplementary Table 8. Second, we demonstrated that chromatin accessibility at the *SATB2* locus was associated with germline UL risk allele status (Figure 6), validating the functionality of these regions in the UL genesis. Third, we showed that the genes mapping to GWAS SNPs (FUMA analysis) were differentially expressed in uterine tissue (ranking first out of 54 examined tissue types in GTEx) and, more strikingly, revealed distinct clusters in RNA-seq data for myometrium and the three tumor subclasses: MED12, HMGA2 and FH (Supplementary Fig. 9). Finally, we emphasize the subclass-specificity of chromatin states seen at GWAS loci (Fig. 5c), which was further exemplified by the *IGF2BP2* GWAS locus (Fig. 5d). Taken together, these results point towards the potential subclass-specificity of multiple GWAS loci, an intriguing question also for future research.

4. Tissue specificity of GWAS loci

It is useful to use analysis tools such as MAGMA and FUMA to verify tissue-specific genetic associations. This may reveal genetic factors that are specific to a particular tissue (e.g.uterus). This will provide clues to the tissue-specific mechanisms of the disease.

We implemented the reviewer's suggestion and included the MAGMA and FUMA analyses of tissue-specificity in the text (lines 367-382). These results strengthened our claims on two fronts: First, the FUMA gene-set analysis showed strong enrichment for biological processes involved in DNA damage and telomere organization, which coincides with our original conclusions about the roles these GWAS loci have in UL tumorigenesis. Second, the FUMA differential expression analysis revealed the strongest enrichment for uterine tissue among the 54 different tissue types examined, supporting the notion that our GWAS signals arise from loci that are specific to this tissue type.

REVIEWERS' COMMENTS

Reviewer #1 (Remarks to the Author):

The authors have sincerely addressed questions/suggestions from both reviewers: These include revision of texts and abstracts, inclusion of statements for clarity, and new figures. The manuscript should be acceptable for publication.

We thank both reviewers for their comments and suggestions for improving the manuscript.

Reviewer #2 (Remarks to the Author):

The authors fully addressed this reviewer's comments in the Revised paper. We believe that this paper is appropriate for publication in nature communications.

We thank both reviewers for their comments and suggestions for improving the manuscript.